# Longitudinal single cell atlas identifies complex temporal relationship between type I interferon response and COVID-19 severity

Quy Xiao Xuan Lin[1,15], Deepa Rajagopalan[1,15], Akshamal M. Gamage[2], Le Min Tan [1], Prasanna Nori Venkatesh [1], Wharton O. Y. Chan [2], Dilip Kumar [3], Ragini Agrawal[4], Yao Chen[5], Siew-Wai Fong[5], Amit Singh [4], Louisa J. Sun[6], Seow-Yen Tan[7], Louis Yi Ann Chai[8,9], Jyoti Somani[8,9], Bernett Lee[3,5,10], Laurent Renia[5,10], Lisa F P Ng [5], Kollengode Ramanathan [9,11], Lin-Fa Wang [2,12], Barnaby Young [10,13,14], David Lye[9,10,13,14], Amit Singhal [3,5,16] ✉ & Shyam Prabhakar [1,16] ✉

Due to the paucity of longitudinal molecular studies of COVID-19, particularly those covering the early stages of infection (Days 1-8 symptom onset), our understanding of host response over the disease course is limited. We perform longitudinal single cell RNA-seq on 286 blood samples from 108 age- and sex-matched COVID-19 patients, including 73 with early samples. We examine discrete cell subtypes and continuous cell states longitudinally, and we identify upregulation of type I IFN-stimulated genes (ISGs) as the predominant early signature of subsequent worsening of symptoms, which we validate in an independent cohort and corroborate by plasma markers. However, ISG expression is dynamic in progressors, spiking early and then rapidly receding to the level of severity-matched non-progressors. In contrast, cross-sectional analysis shows that ISG expression is deficient and IFN suppressors such as *SOCS3* are upregulated in severe and critical COVID-19. We validate the latter in four independent cohorts, and *SOCS3* inhibition reduces SARS-CoV-2 replication in vitro. In summary, we identify complexity in type I IFN response to COVID-19, as well as a potential avenue for host-directed therapy.

The COVID-19 outbreak, caused by the single-stranded RNA virus, Severe acute respiratory syndrome coronavirus-2 (SARS-CoV-2), has resulted in more than half a billion confirmed infections and over 6.2 million deaths globally as of July 2022, and the pandemic is still ongoing. Clinically, COVID-19 causes a broad spectrum of illnesses, ranging from an asymptomatic or mild upper respiratory tract infection to pneumonia, hypoxia, and acute respiratory distress syndrome (ARDS)[1].

Despite extensive research globally, our understanding of COVID-19 disease dynamics remains inadequate, as does our knowledge of markers for COVID-19 patient monitoring and stratification[2]. This is at

least partially attributable to insufficient large-scale longitudinal studies that control for key confounders such as age and disease severity at the time of sampling. In the majority of such studies based on single-cell RNA-seq, the longitudinal trajectory of disease severity is unavailable, and thus one cannot easily infer temporal dynamics[3–8]. Moreover, due to the inherent biases of hospital-based recruitment, studies of SARS-CoV-2 host response have largely focused on later-stage patients (>8 days from symptom onset). Lastly, disease duration (days from symptom onset) at the time of sample collection is unavailable in many cases[6,8–10]. Thus, there is a paucity of well-annotated molecular studies of SARS-CoV-2 host response over the entire disease

course. By limiting our understanding of COVID-19 markers and disease biology, these factors may also have contributed to the insufficiency of therapeutic options.

Since the host immune response to SARS-CoV-2 is a major driver of ARDS and other adverse outcomes, host factors are commonly used as markers for predicting progression to severe COVID-19. For example, immunological studies have identified multiple secreted factors in plasma as prognostic markers of COVID-19 disease outcome, including C-reactive protein (CRP), procalcitonin, lactate dehydrogenase (LDH), D-Dimer, IL-6, IL-10, soluble TNF receptors, CXCL-10, TGFα, IL-16, and IL-23[11–17]. However, these secreted markers do not reflect the properties of the immune cells themselves.

Multiple studies have associated alterations in blood cell proportions with COVID-19 severity, including enrichment of neutrophil precursors, MDSCs, HLA-DR[lo] monocytes and CD169[+] activated monocytes, depletion of HLA-DR[hi]CD11c[hi] inflammatory monocytes and CD4[+], CD8[+] and γδ T cells, T-cell exhaustion and an increased ratio of CD8[+] effector T cells to effector memory T cells[8,13,18–24]. While these (mostly cross-sectional) studies provide insights into disease mechanisms and diagnostic markers, only a minority account for two major confounding factors: disease duration (immune cell phenotypes evolve over the disease course) and age (severe COVID-19 patients are on average substantially older).

In this study, to systematically characterize the longitudinal dynamics of host response and the molecular mechanisms of severe COVID-19, we perform scRNA-seq and immune repertoire profiling on 286 peripheral blood samples collected longitudinally from an age- and sex-matched cohort of 108 patients, and document disease duration and severity at the time of sample collection. 73/108 participants (68%) were sampled at least once during days 1–8. Uniquely, in addition to the conventional clustering of single cells into discrete subtypes, we analyze host response as a continuum of cell states. Data analysis is performed at high temporal resolution (intervals of 4–5 days). This high-resolution continuum approach is important for identifying early upregulation of type I interferon (IFN)-stimulated genes (ISGs) in mild patients, spanning multiple cell subtypes, as the most salient predictor of subsequent increase in disease severity. Remarkably, this IFN signature is highly transient, receding at the very next sampling. We validate this prognostic ISG signature in an independent cohort[18] and further corroborate it by examining plasma cytokine and chemokine levels in the discovery cohort. Intriguingly, cross-sectional analysis reveals the opposite correlation between ISGs and disease severity: *SOCS3* and other IFN-*suppressing* factors were upregulated in severe COVID-19. This suggests a potential mechanism for the well-known attenuation of type I IFN signaling in severe COVID-19[3,25]. Targeting SOCS3 diminishes SARS-CoV-2 replication in vitro. Thus, our study reveals complexity in the temporal dynamics of the host response to COVID-19, provides biological insights, and identifies early markers and potentially targetable mechanisms of severe disease, thereby opening up further avenues for patient monitoring and therapeutics development.

## Results

### Generation and initial characterization of longitudinal single-cell dataset

To comprehensively characterize the dynamic host immune response to SARS-CoV-2 and map the molecular course of disease progression, we collected peripheral blood samples from 112 COVID-19 patients with varying disease severity (108 after QC; Fig. 1A). These samples were collected early in the pandemic (March-June 2020) as part of a larger cohort study of COVID-19 conducted in Singapore[26]. Patients with serial blood samples in this cohort were identified, and individuals with a severe or critical outcome were matched by sex and age (+/−5 years) to patients with asymptomatic or mild outcomes (Supplementary Data 1). 317 blood samples were drawn longitudinally, with at least

two distinct time points sampled per individual, and PBMCs were isolated. PBMC samples were classified into 6 groups based on the severity at the time of sample collection: Asymptomatic, Mild1, Mild2, Moderate, Severe, and Critical (286 samples after QC; Fig. 1B, Supplementary Data 1). Temporally, we grouped samples into four stages based on the number of days since symptom onset (Days 1–4, Days 5–8, Days 9–14, and Day 15 onwards; Fig. 1C). We defined the early-stage of infection as the period preceding clinical deterioration, i.e., Days 1–8[10,27]. The majority of patients who presented as Mild1 (29/37) remained at the same severity; the remaining 8 patients, defined as Mild1 Progressors, mostly progressed to Mild2, i.e., pneumonia, at subsequent time points. Similarly, 6/34 patients who presented as Mild2 subsequently progressed to more severe disease (Mild2 Progressors) and required supplementary oxygen or ICU admission (Fig. 1C). One advantage of our longitudinal study design is the ability to compare gene expression differences at presentation between these Progressors and Non-Progressors, in order to identify cell type-specific prognostic signatures (Fig. 1C).

We performed single-cell analysis on PBMCs in batches of 16 samples each, taking care to minimize batch-to-batch differences in age, sex, severity, and ethnicity (Supplementary Data 2). To reduce technical variation, we pooled the cells from the 16 individuals in each batch before encapsulating cells in droplets. We then performed scRNA-seq and TCR- and BCR-seq using the 5′ v2 Immune Profiling protocol from 10X Genomics. In parallel, we genotyped each sample using the Illumina GSA-MD-v3 chip and used reads covering genetically polymorphic sites to bioinformatically demultiplex the scRNA-seq data, i.e. to assign cells to their sample of origin[28]. After discarding doublets and low-quality cells (Methods), we obtained 346,680 high-quality cells from 286 samples representing 108 COVID-19 patients. We used the Reference Component Analysis (RCA2) algorithm[29,30] to perform supervised clustering on these cells and thereby grouped them into seven major cell types: T, NK, B, plasma B, monocyte, conventional dendritic cell (cDC), and platelet (Fig. 1D, Supplementary Fig. 1, Supplementary Fig. 2A). Intriguingly, monocytes from critical patients showed a distinctive expression phenotype (Fig. 1E, red cells), suggesting that single-cell transcriptomics could help uncover subtle shifts in cell state within the major cell types. We therefore used de novo (unsupervised) clustering to further assign T cells, NK cells, B cells, and myeloid cells to their respective subtypes (Fig. 1F–H and Supplementary Fig. 2B–D).

Having defined the cell type landscape of COVID-19 PBMC samples, we investigated the possibility of clonal expansion of T cells in response to SARS-CoV-2 infection. In the extreme scenario of uniform clonal abundance and zero sampling noise, the top 5 clones would be no more abundant than the next 45 clones. Consequently, the former would account for 10% (5/50) of the cells harboring the 50 most abundant clones. Indeed, the latter proportion, which we define as the clonality index, did not substantially exceed this baseline expectation of 10% in critical patients beyond Day 8, suggesting minimal clonal expansion in this group (Fig. 1I and Supplementary Fig. 3A). A similar pattern was observed during Days 1–8, though the effect was not significant, perhaps due to the lower number of severe and critical samples from the first 8 days (Supplementary Fig. 3B). BCR clonality showed no clear trend (Supplementary Fig. 4A–D), potentially due to the lower number of BCR sequences identified (Supplementary Fig. 5). Overall, we observed clonally expanded T cells mainly in CD8[+] T cells (Supplementary Fig. 3C, D), and T-cell clonality was significantly negatively correlated with COVID-19 severity (Fig. 1I), suggesting a defective T-cell response in severe and critical patients[31].

### Early elevation of IFN-stimulated gene expression predicts subsequent increase in severity

To identify early prognostic markers of progression to severe disease, we first classified patients into two groups (Fig. 1C): those whose

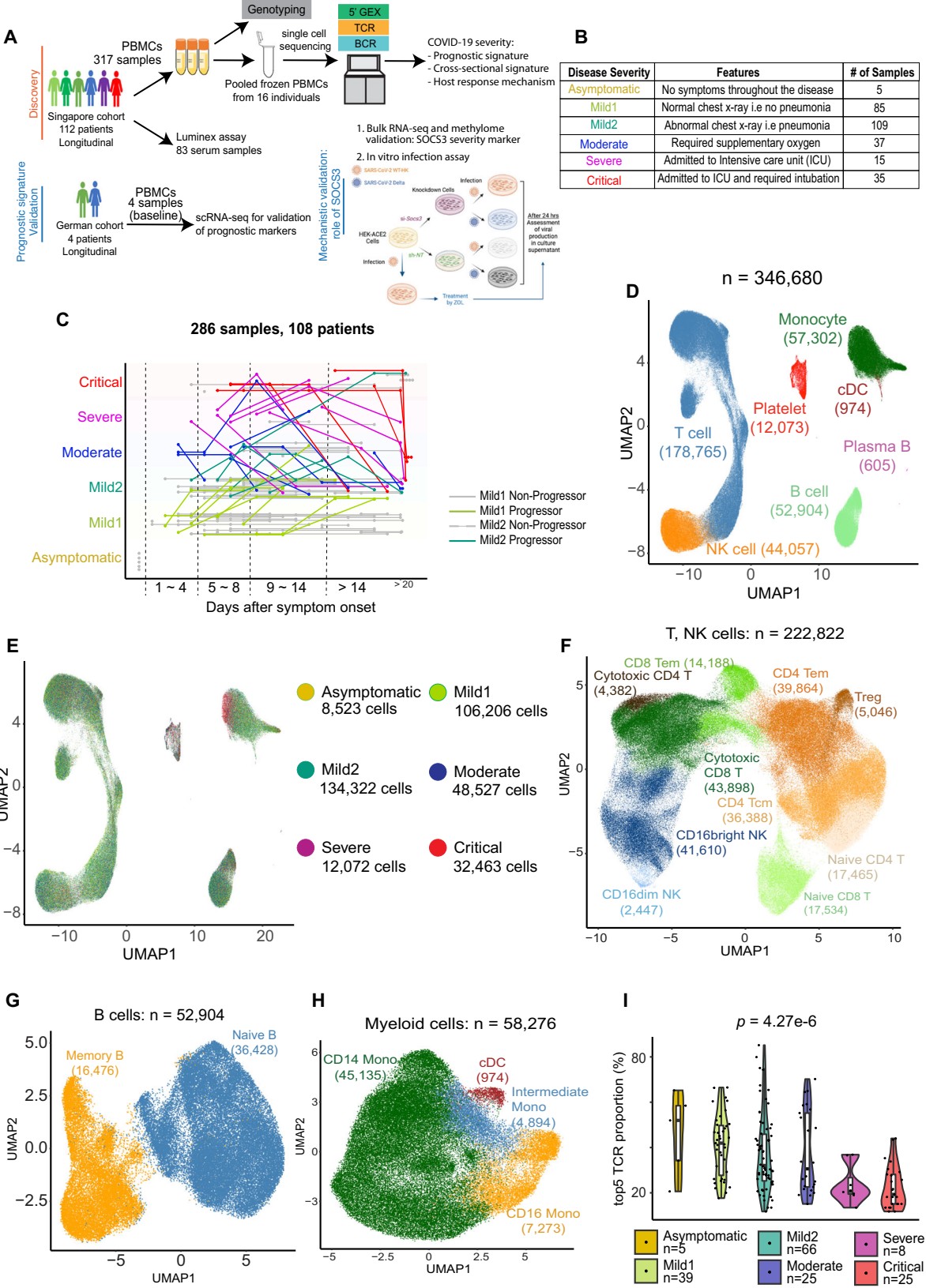

symptoms were stable across the longitudinal disease course (non-progressors) and those who progressed to more severe disease after the baseline sample taken at presentation during Days 1–8 (Progressors). For patients who presented as Mild1, we identified genes that were differentially expressed (DE) in baseline samples from Progressors, relative to Non-Progressors sampled at the same disease

duration. For each major cell type (T/NK, B, myeloid), we defined the set of DE genes as the union of DE genes for the corresponding cell subtypes (Methods). Remarkably, in all three major cell types, the genes upregulated during the early stages of the disease course in Mild1 Progressors were most strongly enriched for type I IFN signaling (FDR: *Q* < 1e-20; Fig. 2; Supplementary Data 3 and 4). Type I IFN

**Fig. 1 | Longitudinal single-cell workflow and landscape of PBMC gene expression. A.** Schematic representation of the data generation workflow in the study. **B.** Disease severity definitions for categorizing each sample by clinical parameters at the time of sample collection. A number of samples under each disease severity is indicated. **C.** Longitudinal disease course for 108 patients (after QC). Dots represent the collection day and disease severity of each of the 286 samples (after QC). Gray lines: the trajectory of individuals with stable disease. Colored lines: individuals with variable severity, colored by the severity of the at-presentation sample. **D.** Uniform manifold approximation and projection (UMAP) reduced-dimensionality representation of 346,680 high-quality post-QC single cells in gene expression space, annotated with major cluster labels using supervised clustering by the Reference Component Analysis 2 (RCA2) algorithm. **E.** Same as **D**, colored by disease severity of each sample. **F-H.** UMAP plots of T, NK cells (**F**), B cells (**G**), and myeloid cells (**H**), with cells colored by subtype based on unsupervised clustering of single-cell transcriptomes. **I.** T-cell clonality index (fraction of T cells derived from the 5 most abundant TCR clones; beyond Day 8), estimated using the single-cell immune profiling assay. *p*-value: Kruskal–Wallis test. Source data are provided as a Source Data file. Box-and-whisker plots show the median (center line), 25th, and 75th percentile (lower and upper boundary), with 1.5x interquartile range indicated by whiskers and outliers shown as individual data points.

signaling was also the most enriched annotation among genes upregulated at baseline (Days 5–8) in Mild2 Progressors (Supplementary Fig. 6 and Supplementary Data 3, 4). We did not test for prognostic signatures in Mild2 Progressors during Days 1–4, or in Moderate Progressors, since the number of patients in these categories was too small. This association of type I IFN upregulation with worse prognosis was surprising, since type I IFNs suppress viral replication and constitute a primary component of the innate immune response to viral infection[32–36]. Moreover, previous studies showed a negative correlation between IFN signaling and COVID-19 severity, i.e., the opposite trend[34–36]. However, these studies were mostly cross-sectional rather than longitudinal, and also based on later-stage samples (>8 days postsymptom onset). Our results thus indicate that the temporal dynamics of IFN response to SARS-CoV-2 are more complex than previously reported.

In all prognostic comparisons for Mild1 and Mild2 patients, across all three major cell types, a core group of 13 ISGs was upregulated in Progressors: *ISG15, ISG20, IFIT1, IFIT2, IFIT3, IFITM1, IFI6, IFI35, BST2, RSAD2, IRF7, MX1* and *MX2* (Fig. 3A and Supplementary Data 3). Consistently, the cell states (locations in gene expression space) enriched in Progressors (Fig. 2) precisely matched those associated with high expression of these ISGs (UMAP plots in Fig. 3B–D). We therefore defined the average normalized expression of these 13 genes at baseline as a prognostic score; this score was significantly elevated in Progressors (Fig. 3B–D, box-plots; Methods). This prognostic signature showed no significant association with the two most prevalent comorbidities, acute myocardial infarction, and diabetes mellitus, or with age and sex (Supplementary Fig. 7A). The latter result could be attributable to the fact that our cohort was selected to minimize systematic differences in age and sex (Supplementary Fig. 7B, C). Receiver operating characteristic-area under curve (ROC-AUC) values for our prognostic score ranged from 0.81-0.84 (Fig. 3B–D), suggesting that upregulation of these ISGs constitutes a robust early-stage signature of subsequently increased disease severity.

Although the vast majority of published transcriptomic analyses of the host response to SARS-CoV-2 were unsuitable for validating our prognostic signature, we were able to identify five individuals from a German cohort[18] with single-cell PBMC expression data that satisfied the necessary criteria, namely early sample collection (Days 1–8), knowledge of disease duration, longitudinal tracking of COVID-19 disease severity, and mild symptoms at baseline. In baseline samples from these five individuals, we averaged the expression of each of the 13 genes comprising our prognostic signature across all single cells of all types in any given sample, based on which we calculated a single prognostic signature score for each sample (Methods). Despite the small cohort size, the prognostic score of Progressors in this cohort was significantly higher than that of non-Progressors (*p*-value = 0.012, Student's *t*-test; Fig. 3E). Moreover, when examined individually, 8/13 genes showed significantly higher expression in Progressors (*p*-value ≤ 0.05, Student's *t*-test). These results validate our prognostic ISG signature in an independent cohort.

We next correlated our prognostic signature with inflammatory cytokine and chemokine levels in plasma. We obtained data from 48 individuals in our cohort with mild disease at baseline (Luminex assay, 83 samples, 11 Progressors, 37 non-Progressors). Across the 83 samples, 3/33 plasma protein markers (IFN-alpha, MCP-1, IP-10) showed significant correlation with the 13-gene prognostic mRNA signature (FDR *Q*-value < 0.01, Fig. 3F). It is thus possible that the upregulation of ISGs in PBMCs from Progressors may represent a transcriptional response to increased IFN-alpha in the plasma. The correlation with plasma MCP-1 (CCL2) and IP-10 is intriguing since these two markers are associated with COVID-19 disease severity[13,37]. Consistently with our results from prognostic scRNA-seq analysis, the average plasma expression scores of these three markers were significantly higher in baseline samples from the 10 Progressors than from the 37 non-Progressors (*p*-value = 0.0089, Fig. 3G, Supplementary Fig. 8). Moreover, all three markers independently showed prognostic power (*p*-value < 0.05, Fig. 3G, Supplementary Data 3). Thus, results from an independent protein-based assay are concordant with the prognostic signature we identified using scRNA-seq, and they again deviate from the prior expectation[32,38] that type I IFN expression is necessarily protective.

In summary, by longitudinally characterizing disease severity and comparing gene and protein expression profiles at baseline, we identified and validated early (Days 1–8) upregulation of ISGs as a prognostic signature that predicts subsequent increase in COVID-19 severity.

## Global, progressive decrease in type I IFN signaling over time

After identifying the prognostic signature of increased severity, our next goal was to characterize the longitudinal evolution of PBMC cell states over time. We first examined temporal shifts in the distribution of T and NK cell states in Mild1 Non-Progressors. We defined the location of each such cell in gene expression space as a cell state, and then examined the 300 cells nearest to it. Within these 300 neighbors, we quantified the fold-enrichment of cells from any particular temporal stage relative to cells from the first temporal (Days 1–4, Methods). To identify cell states that followed a similar trajectory of enrichment or depletion, we then clustered cells by their fold-enrichment over time. Notably, the majority of T and NK cell states in Non-Progressors were either progressively enriched or progressively depleted over time, while a small minority exhibited more complex trajectories (Supplementary Figs. 9–14). Strikingly, cell states depleted over time in Mild1 Non-Progressors (T and NK cells) almost exactly matched those that expressed ISGs (Fig. 4A). Indeed, type I IFN response was the most highly enriched functional annotation among markers of these depleted cell states (Fig. 4B). Moreover, this group of cells was progressively depleted over time even in the other severity categories (Fig. 4A, B). Furthermore, exactly the same trend was observed for B and myeloid cells (Fig. 4C–F). In contrast, we did not observe strong enrichment of any particular functional category among cell states that were progressively enriched over time (orange cells in Fig. 4A, C, E; Supplementary Figs. 15–17, Supplementary Data 4). Lastly, cell states depleted over time in T, NK, B and myeloid cells of Mild1 and Mild2 Progressors again showed the greatest enrichment for interferon responsive genes, including interferon-stimulated genes (*ISG15, ISG20, IFIT1, IFIT2, MX1*), interferon regulatory factors (*IRF3*,

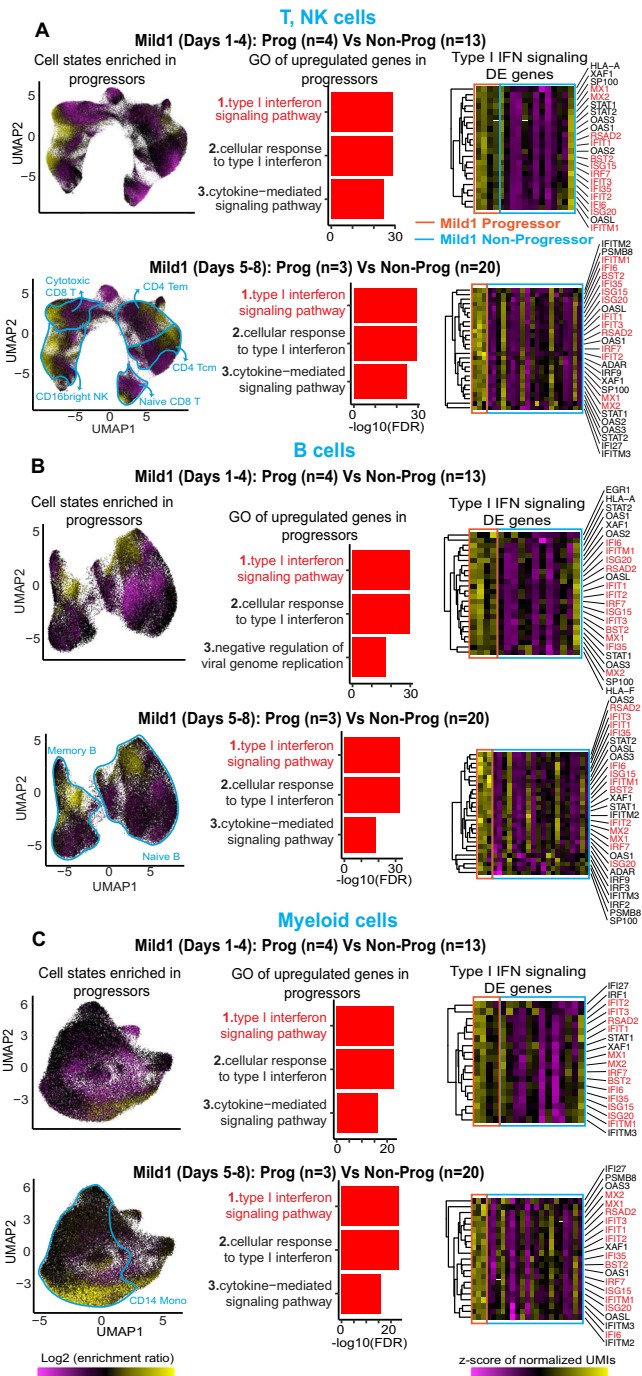

**Fig. 2 | Comparison of Progressors and Non-Progressors to identify a COVID-19 prognostic signature.** UMAP representations show the cell-state enrichment fold-change between Mild1 progressors and non-progressors in T, NK cells (**A**), B cells (**B**), and myeloid cells (**C**). Yellow: cell states (gene expression neighborhoods) enriched in Progressors. Magenta: cell states depleted in Progressors. Red bar plots: enriched Gene Ontology (GO) terms associated with the union across all cell sub-types of genes upregulated in Progressors. Heatmaps on the right: pseudobulk expression levels of type I IFN genes upregulated in Progressors. Source data are provided as a Source Data file.

*IRF7*, *IRF9*) and 2′-5′-oligoadenylate synthetases (*OAS1*, *OAS2*, *OASL*) (Supplementary Figs. 18–20; Supplementary Data 3, 4). These results from PBMC single-cell transcriptomics are consistent with the decline over time in plasma IFN-alpha protein levels (Supplementary Fig. 8). In summary, the predominant trend in the three major PBMC cell types from Progressors and Non-Progressors was a gradual reduction over

time in the number of cells expressing ISGs, indicating a steady, global decrease in innate antiviral response.

Since type I IFN signaling was the dominant, unifying theme both as an early prognostic signature and as a driver of subsequent temporal dynamics, we defined a broad metagene comprising all 13 genes related to our ISG score and quantified its average (pseudobulk) expression in each major cell type of each sample. This analysis confirmed once again that Progressors showed elevated ISG expression at presentation, relative to stage and severity-matched Non-Progressors (Fig. 4G–I). Interestingly, once these individuals progressed to the next level of severity, their 13-gene ISG score was no longer atypical. Rather, it matched the levels seen in matched Non-Progressors. Thus, the prognostic type I IFN response signature was transient, and observable only in baseline samples collected during Days 1–8, before the peak of disease severity. By Days 9–14, type I IFN response receded to levels close to those seen in asymptomatic cases regardless of severity, and beyond Day 14 the reversion to asymptomatic levels was complete (Fig. 4G–I, dashed line). Consistently, longitudinal analysis of plasma IFN-alpha levels shows high expression at baseline in Progressors, followed by a drop at the very next timepoint (Supplementary Fig. 8).

## Cross-sectional analysis: immune cell type and subtype proportions vary with severity

A major advantage of unbiased scRNA-seq analysis is that we can test for associations between PBMC cell type composition and COVID-19 severity. Since severe and critical cases were primarily sampled beyond Day 8, we initially examined the relative proportions of major cell types in samples from Day 9 onwards. As previously reported[3,9,39–41], T-cell proportions decreased with increasing severity, which is consistent with the hallmark lymphopenia widely reported in severe patients[18] (Fig. 5A and Supplementary Fig. 21). However, this trend did not apply to all lymphocytes: B-cell proportions showed no consistent correlation with severity, and NK cells were depleted only in critical samples. We observed that cDCs were strongly depleted with increasing severity, while monocytes and platelets were strongly increased. Within each of the major immune cell types (T and NK, B, myeloid), we further examined associations between subtype proportions and severity (Fig. 5B, C and Supplementary Fig. 22). B-cell subtype proportions showed only weak associations with severity. Cytotoxic CD8⁺ T cells and CD8⁺ T effector memory cells were depleted in severe samples (relative to all T, NK cells), as were intermediate and CD16⁺ monocytes. In contrast, CD14⁺ monocytes were strongly enriched with increasing severity. The above observations are consistent with previous cross-sectional studies of COVID-19[42,43], though we are not aware of any single study that detected all of the associations visible in our single-cell data. Intriguingly, in addition to the above, we noticed a significant increase in CD4⁺ central memory T cells (Tcm) with increasing severity (*p*-value = 7.62e-5). These cells express high levels of *CD69*, a well-known T-cell activation marker, suggesting that CD4⁺ Tcm cells were stimulated to confer systemic protection following their encounter with cognate viral antigens. However, the above-mentioned depletion of cytotoxic and effector memory CD8⁺ T cells with increasing severity suggests a dysfunctional break in the link between T-cell activation and effector function in severe COVID-19[31].

## Cross-sectional cell state analysis: suppressors of IFN signaling are enriched in severe COVID-19

We hypothesized that the numerous differences in immune cell type and subtype abundance observed between mild and severe COVID-19 (Fig. 5) may not be independent. Rather, they could reflect differential signaling traits shared across multiple cell types. We therefore used a cross-sectional equivalent of the longitudinal cell state enrichment analysis described above (Fig. 4A, C, E; Methods). In this case, we used the same 300-cell neighborhoods to calculate the fold-enrichment of each cell state in Mild2, Moderate, Severe, and Critical cases relative to

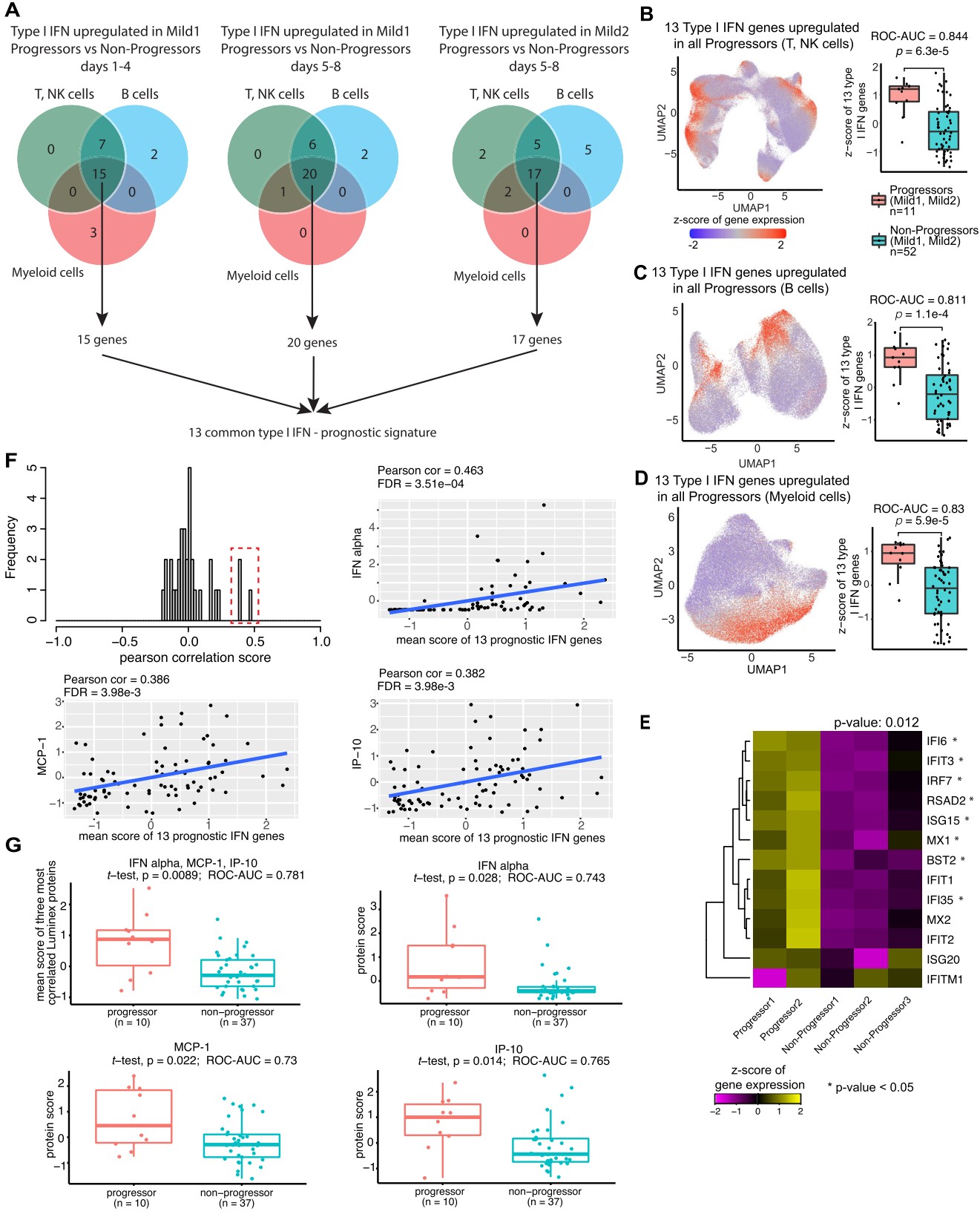

Mild1 (Supplementary Figs. 23–26). To distinguish between the effects of disease severity and duration, we explicitly controlled for the latter as a confounder. We performed this analysis on T and NK cells, B cells, and Monocytes separately (Methods). Within each of these three major cell subsets, individual cell states were clustered by their fold-enrichment trajectories to define states whose abundance

progressively increased or decreased in abundance as disease severity increased (Fig. 6A, D, G; Supplementary Figs. 23A, B; 25A, B; 26A, B).

Although subtype-level abundance analysis indicated that CD4⁺ Tcm cells as a whole were enriched with increasing disease severity (Fig. 5B), the higher-resolution analysis described above revealed a more complex scenario. Intriguingly, only one subpopulation within

**Fig. 3 | Identification of 13 type I IFN signaling gene prognostic signatures. A.** Venn diagrams show the overlapping of type I IFN signaling genes upregulated in Progressors compared to Non-Progressors across three cell types. In the end, 13 type I IFN signaling genes were identified as prognostic marker genes. **B–D** Metagene z-score (normalized relative to all samples from Days 1–4) of 13 type I IFN signaling genes in T, NK cells, B cells, and myeloid cells. Left panels: scRNA-seq UMAP plot colored by metagene z-score. Right panels: box-plots of the pseudobulk (averaged across cells in one sample) metagene z-score in Progressors ($n = 11$) vs. Non-Progressors ($n = 52$). p-value: Student's t-test (one-sided). Receiver operating characteristic−area under curve (ROC-AUC) values show the accuracy of the 13-gene prognostic signature for predicting progression to more severe disease. **E** Heatmap shows the scaled pseudobulk expression levels (expression z-score) of the 13 prognostic ISGs in baseline samples from 2 Mild Progressors and 3 Mild non-Progressors in the German cohort[94] *: p-value ≤ 0.05, Student's t-test (one-sided) for differential expression between Progressors and non-Progressors. p-value is shown

above heatmap: the difference between Progressors and non-Progressors in the trimmed mean of the 13 genes (greater). **F** Histogram: expression correlation of 41 plasma proteins with the prognostic 13-gene ISG signature across 83 samples. The red box indicates the three highly correlated proteins: IFN-alpha, MCP-1, and IP-10. Scatterplots illustrate the correlation between the three proteins and the prognostic 13-gene signature. Each dot represents a single sample. Blue lines: linear regression. FDR: Q-values: Benjamini Hochberg correction on linear regression p-values. **G** The first box plot shows the average of the protein expression scores of IFN-alpha, MCP-1, and IP-10 in baseline samples from 10 Mild Progressors (red) and 37 Mild non-Progressors (blue). The rest three box-plots: are individual plasma markers. p-values: Student's t-test (one-sided). Box-and-whisker plots show the median (center line), 25th, and 75th percentile (lower and upper boundary), with 1.5x inter-quartile range indicated by whiskers and outliers shown as individual data points. Source data are provided as a Source Data file.

CD4+ Tcm cells was enriched with increasing severity (Fig. 6A, orange cells), and in fact some other subpopulations showed the opposite trend (blue cells, depleted). The enriched subpopulation within CD4+ Tcm showed significant upregulation of genes involved in response to cytokine stimulus and suppression of JAK-STAT signaling (Fig. 6B and Supplementary Data 5, 6). Specifically, upregulated genes included *SOCS3, SOCS2, ITGA6*, and *PRDM1*, all of which have been implicated in suppressing the response to type I IFNs (Fig. 6B). Consistently, CD4+ Tcm cells expressing ISGs were depleted with increasing severity (Figs. 4A and 6A; Supplementary Fig. 23). This mechanism was not unique to CD4+ Tcm cells. Rather, subpopulations expressing an overlapping set of suppressors of type 1 IFN signaling including *SOCS3* were enriched in multiple other T and NK cell subtypes: CD4+ Tem, naive CD4+, and naive CD8+ T cells (Supplementary Fig. 24 and Supplementary Data 5, 6). Consistently, these subtypes were also depleted for ISG-expressing cells. Overall, *SOCS3* expression in T and NK cells increases (cross-sectionally) with COVID-19 severity at all disease durations, though the increase is most pronounced beyond Day 14 (Supplementary Fig. 27A and Supplementary Data 7). In contrast to this cross-sectional trend, *SOCS3* expression during Days 1–8 did not appear to be prognostic for disease progression (Supplementary Fig. 27B). Lastly, ISG-expressing cells within the cytotoxic CD8+ T-cell population were strongly depleted with increasing severity (Fig. 6C). In addition, all T and NK subpopulations enriched with increasing severity showed high expression of negative regulators of apoptosis (Supplementary Data 6), including various subsets of the following genes: *BCL2, BCL3, PIM1, PIM2, MYC, DDIT4* and *XBP1*. In summary, high-resolution differential cell state analysis within T and NK cells revealed multiple *SOCS3*-high subpopulations that may contribute to the suppression of type I IFN response observed in severe COVID-19. The same subpopulations also displayed anti-apoptotic signatures, which could potentially be related to suppression of type I IFN signaling[44,45].

B cells were also systematically altered in severe and critical COVID-19. Among memory B cells enriched with increasing disease severity, we observed upregulation of genes involved in neutrophil-mediated immunity, including *TNFRSF1B, FCRL3, FCRL5*, and *FGR* (Fig. 6D, E and Supplementary Fig. 25). This suggests that memory B cells are primed for maturation in severe COVID-19 and could facilitate adaptive immunity upon reinfection[35,46]. Among naive B cells, we found the same pattern described above, namely consistent depletion of the ISG-expressing subpopulation (Fig. 6D, F). In summary, the above results suggest a broadly shared defect in immune response in lymphoid cells.

Myeloid (monocyte and cDC) cells showed the most dramatic shifts in cell state: virtually every subregion in gene expression space was either enriched or depleted with increasing severity (Fig. 6G and Supplementary Fig. 26). Although ISG-expressing myeloid cells (Fig. 4E) were consistently depleted, many other cell states were also less abundant in severe COVID-19. In particular, CD14+ monocytes

showed a striking dichotomy. Within this subtype, the subpopulation expressing effectors of neutrophil-mediated immunity and neutrophil degranulation was uniformly enriched (Fig. 6H). These cells expressed *S100A8, S100A9*, and *S100A12*, all of which are implicated in cytokine storms observed in severe COVID-19[47]. In contrast, most of the remaining CD14+ monocyte subpopulations were depleted with increasing severity. Markers of depleted cells were associated with antigen presentation and processing via MHC class II (Fig. 6I). The latter subpopulation performs the central function of bridging the innate and adaptive immune responses. Substantial depletion of these cells indicates defective myelopoiesis, which could contribute to increased disease severity[14,18,48].

## *SOCS3* and severe COVID-19

To further investigate the role of *SOCS3* in severe COVID-19, we examined the expression of this gene using PBMC bulk RNA-seq data from an independent Singapore cohort ($N = 37$; D.K. et al.[49], unpublished observations). In this dataset, *SOCS3* expression was lowest in healthy individuals and increased steadily with increasing disease severity (Fig. 7A; p-value = 0.02, one-way ANOVA). Similarly, *SOCS3* was upregulated in PBMCs of adults with severe COVID-19 in a UK cohort[50] (pseudobulk expression, $N = 37$), and reverted to baseline upon convalescence (Fig. 7B; p-value = 0.018). This effect was not restricted to peripheral blood. *SOCS3* was also upregulated in airway samples of donors with severe disease in the same cohort ($N = 22$; Fig. 7C; p-value = 0.047), as well as nasal swab samples from a US cohort[51] ($N = 51$; Fig. 7D; p-value = 0.013). To examine potential epigenetic factors contributing to SOCS3 upregulation, we analyzed whole-genome bisulfite sequencing data from the Singapore cohort (D.K. et al.[49], unpublished observations) and identified 11 differentially methylated regions at the *SOCS3* locus between severe and healthy PBMC samples (Fig. 7E). All 11 showed decreased methylation in severe donors (Fig. 7F, G), suggesting that reduced DNA methylation near *SOCS3* may contribute to upregulation of the gene in patients with severe COVID-19. In summary, *SOCS3* upregulation is a consistent feature of severe and critical COVID-19, across multiple cohorts, both in peripheral blood and at sites of infection.

To assess the role of *SOCS3* in viral infection, we devised an ex vivo assay in which this gene was silenced by shRNA in HEK293T-Angiotensin-Converting Enzyme 2 (HEK-ACE2) epithelial cells, followed by infection with two strains of SARS-CoV-2 (Fig. 7H). Since airway epithelial cells are a major site of infection, this epithelial cell line, which expresses both the receptors used for SARS-CoV-2 entry, is commonly used as a model system[52]. The average silencing efficiency was 60-80% (Supplementary Fig. 27C). Viral replication was monitored by qRT-PCR quantification of the viral N-gene in the culture supernatant. In knocked-down cells, the quantity of released virus was reduced by orders of magnitude when SOCS3 was silenced (Fig. 7I; p-value = 0.0019, Kruskal–Wallis test), indicating that silencing of *SOCS3* strongly suppressed replication of SARS-CoV-2. We also interfered

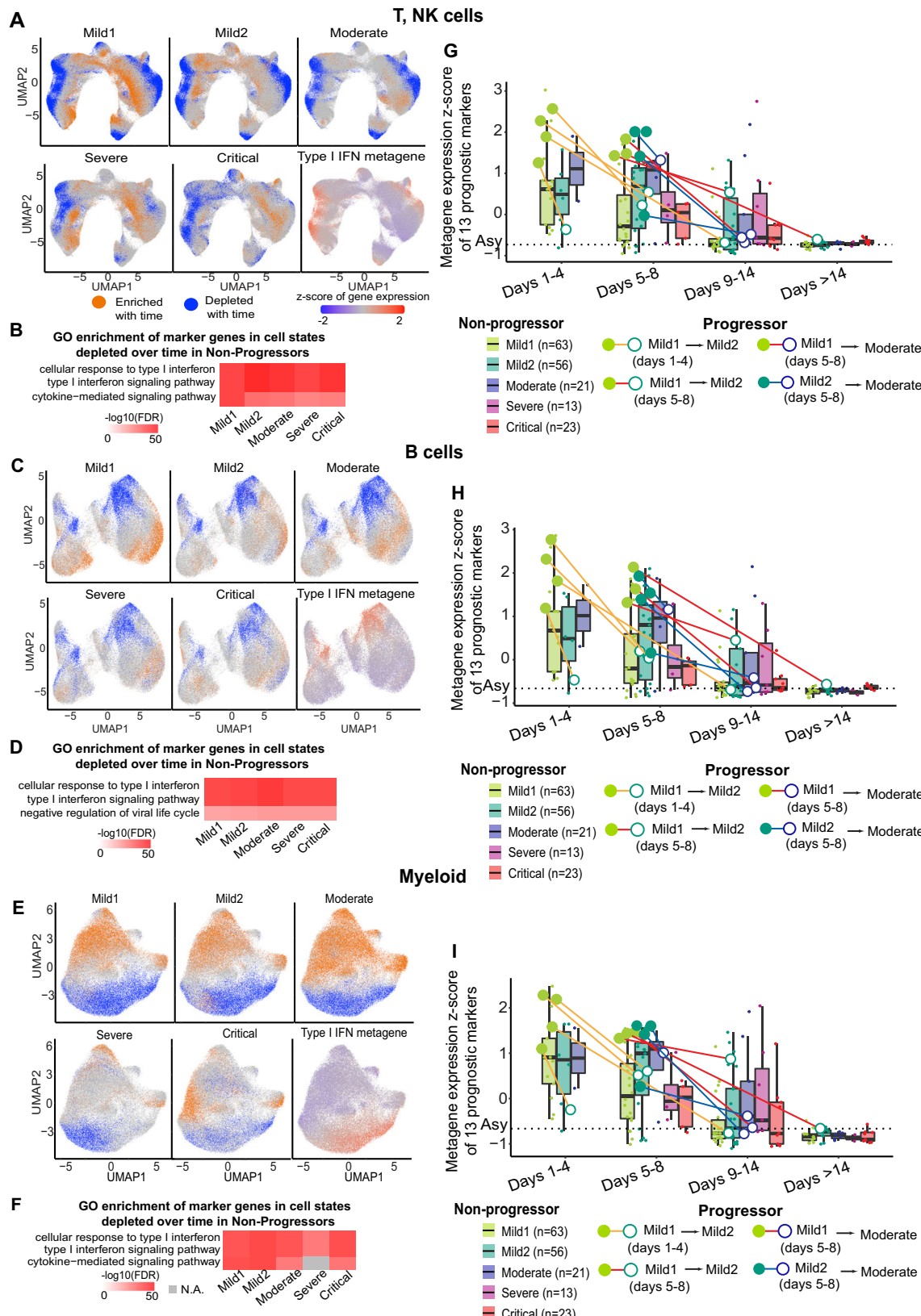

with SOCS3 function by treating SARS-CoV-2-infected HEK-ACE2 cells with the small-molecule inhibitor zoledronic acid (ZOL)[53], a drug used to treat osteoporosis. We again observed a significant decrease in viral replication (Fig. 7H, J; *p*-value = 0.017, *t*-test). Taken together, the above results suggest that the upregulation of *SOCS3* could potentially contribute to COVID-19 disease severity.

## Discussion

We have leveraged scRNA-seq data from 286 PBMC samples from 108 COVID-19 patients to identify the upregulation of ISGs (Supplementary Data 3) as the major early-stage prognostic signature of disease progression, and show that it is remarkably short-lived (Figs. 2–4). Importantly, our single-cell dataset combined two essential

**Fig. 4 | Longitudinal analysis of COVID-19 single-cell transcriptomes. A** UMAP representation of cell states (locations of cells in gene expression space) enriched or depleted over time in T, NK cells. A cell is colored orange if the proportion of Non-Progressor Mild1 cells (for example, top-left UMAP) in its immediate vicinity (300 nearest neighbors) increases over time, and blue if the proportion decreases. For comparison, the average scaled expression of all expressed genes related to type I IFN signaling is shown (type I IFN metagene). **B** Top 3 enriched GO terms of marker genes of cell states depleted over time in Non-Progressors (blue cells). **C, D** B cells, same as (**A, B**). **E, F** Myeloid cells, same as (**A, B**). **G–I** Prognostic type I IFN metagene z-score (normalized across all samples) of Mild Progressor (circles) and Non-Progressor (box-plots) samples, grouped by disease severity and duration. Horizontal dashed line: average type I IFN metagene z-score of the 5 asymptomatic (Asy) samples. In each PBMC sample, metagene z-scores were averaged across all T, NK cells (**G**), B cells (**H**), and myeloid cells (**I**). Filled circles: baseline samples of Progressors. Empty circles: second samples of Progressors. Box-and-whisker plots show the median (center line), 25th, and 75th percentile (lower and upper boundary), with a 1.5x inter-quartile range indicated by whiskers and outliers shown as individual data points. Source data are provided as a Source Data file.

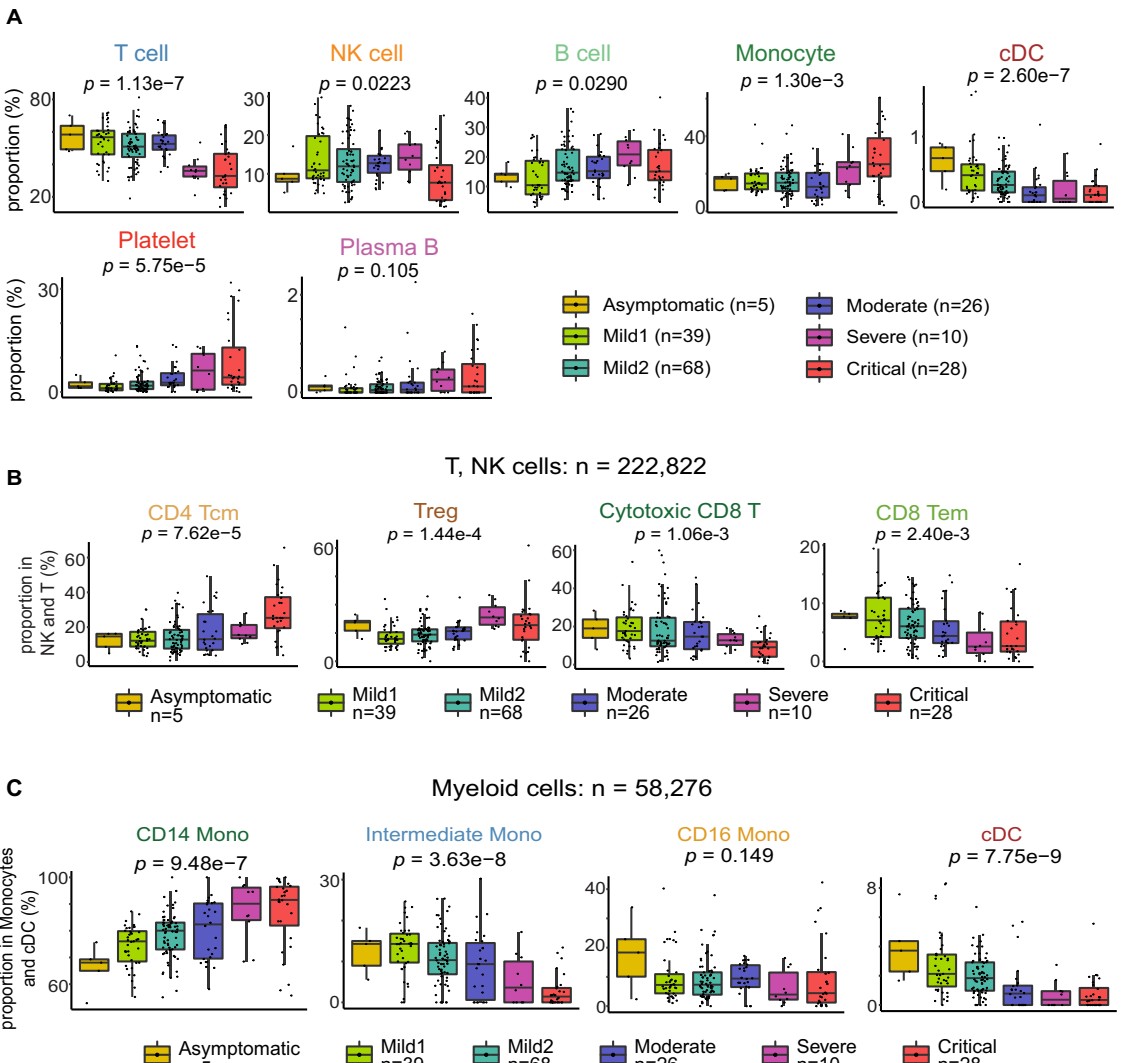

**Fig. 5 | Cross-sectional analysis: the relationship between immune cell type proportions and COVID-19 disease severity. A** Box-plots indicate major cell type proportions (%) in PMBC as a function of disease severity. Each dot represents a single sample. To minimize the confounding effect of disease duration, only samples from Day 9 disease onset onwards are used in this analysis. *p*-values: Kruskal–Wallis test (non-parametric one-way ANOVA for each cell type). **B** Similar to **A**, indicating subtype proportions as a fraction of all T and NK cells. **C** Similar to **A**, indicating subtype proportions as a fraction of all myeloid PBMC (monocytes and cDCs). Box-and-whisker plots show the median (center line), 25th, and 75th percentile (lower and upper boundary), with 1.5x inter-quartile range indicated by whiskers and outliers shown as individual data points. Source data are provided as a Source Data file.

attributes for accurate characterization of the dynamics of host response (Fig. 1A–C): (1) dense longitudinal sampling of early disease stages (Days 1-8) and 2) knowledge of disease severity at the time of sample collection. Mild1 (Days 1–8) and Mild2 (Days 5–8) patients with elevated ISG expression were at greater risk for adverse outcomes, ranging from pneumonia to hypoxia (Figs. 1C, 2, and 3). Since this predominant cellular prognostic signature was shared across the three major cell types in peripheral blood, it could potentially form the basis for a bulk-sample blood test. However, further studies in larger cohorts are needed to examine the clinical utility of this signature, particularly for predicting progression to severe and critical COVID-19. Analysis in larger cohorts would also provide the opportunity to identify the optimal subset of the 13 markers for clinical translation.

The peripheral type I IFN response to COVID-19 was highly dynamic. Regardless of disease severity, ISG expression steadily decreased over time among Progressors and Non-Progressors (Fig. 4), dropping to the level of asymptomatic individuals by Day 14.

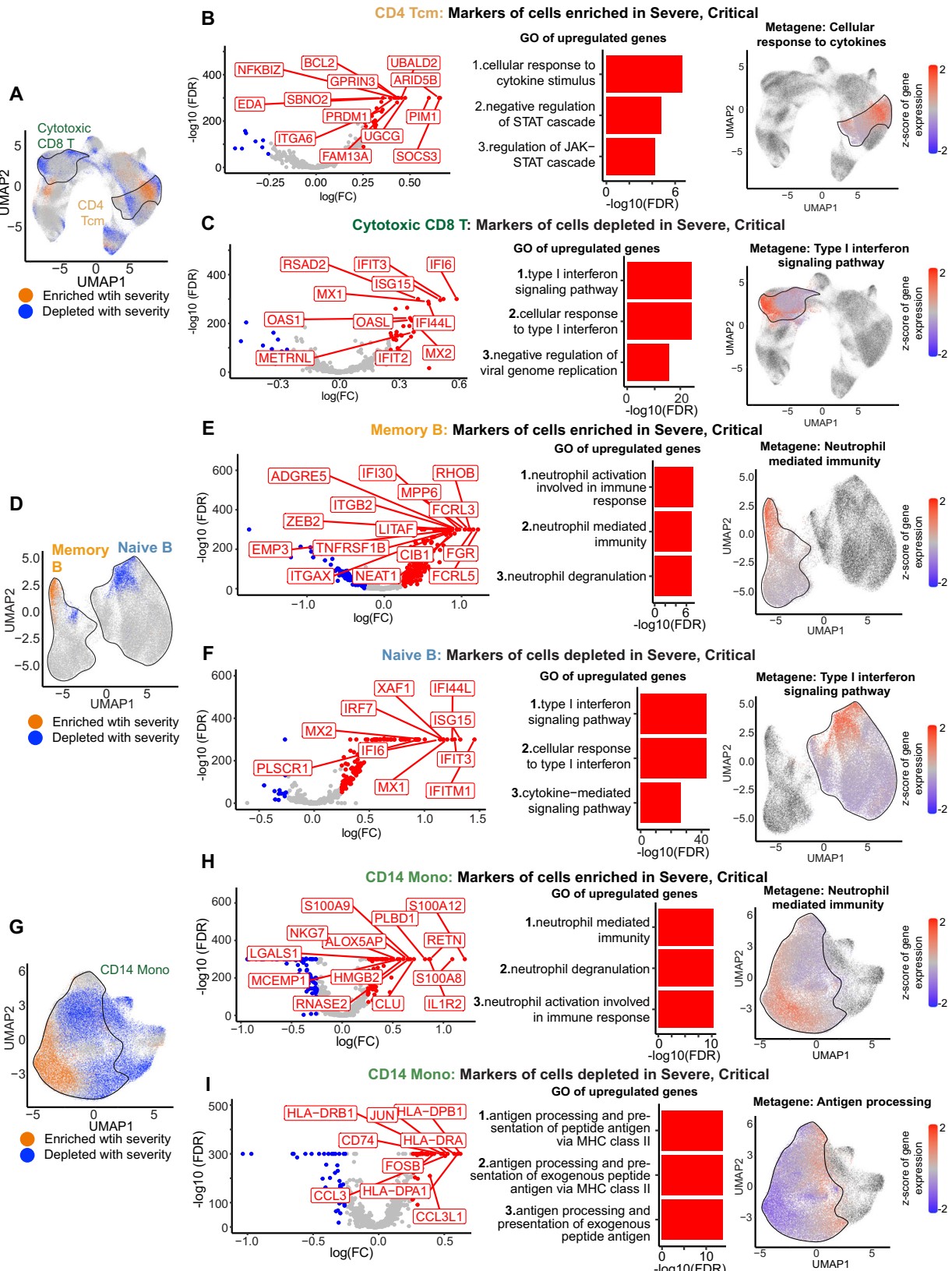

Moreover, although Progressors displayed elevated ISG expression at baseline, they reverted to the levels of Non-Progressors at the very next sampling. Thus, ISG expression subsided before disease severity increased. These findings underscore the complex and dynamic temporal relationship between IFN signaling and COVID-19 disease outcome.

The role of type I IFNs in COVID-19 pathophysiology is controversial. Multiple *cross-sectional* studies have reported deficient type I IFN levels in the peripheral blood of severe patients[34,54,55]. Our cross-sectional analyses confirm that severe and critical patients have low ISG expression in multiple cell subtypes relative to moderate patients (Fig. 6). Since type I IFNs control viral replication and enhance T-cell-

**Fig. 6 | Cross-sectional cell state analysis to identify immune aberrations in severe COVID-19. A** UMAP representation of cell states (locations of cells in gene expression space) enriched or depleted with increasing COVID-19 severity. A cell is colored orange if the proportion of cells in its immediate vicinity (300 nearest neighbors) increases with increasing severity, and blue if the proportion decreases. Black outlines demarcate specific cell subtypes (clusters). **B** Left: Volcano plot of marker genes of CD4[+] Tcm cell states enriched with increasing severity (orange Tcm cells vs all other Tcm cells). Middle: the top 3 corresponding enriched GO terms. Right: UMAP plot, each cell is colored by the average of the standardized gene expression values of the marker genes belonging to the most enriched GO term. **C** Same as **B**, for Cytotoxic CD8[+] T-cell states that are depleted with increasing disease severity. **D** Same as **A**, for B cells. **E** Same as **B**, for memory B-cell states enriched with increasing severity. **F** Same as **B**, for naive B-cell states, depleted with severity. **G** Same as **A**, for myeloid cells. **H** Same as **B**, for CD14[+] monocyte cell states enriched with increasing severity. **I** Same as B, for CD14[+] monocyte cell states depleted with severity. Source data are provided as a Source Data file.

mediated adaptive responses[56], this suppression of ISGs in severe COVID-19 is highly plausible. Consistently, genetic and autoantibody analyses have linked severe COVID-19 with deficient type I IFN signaling[57]. Thus, it has been suggested that reduced type I IFN signaling may increase disease severity by compromising the antiviral host response. Despite these conclusions from *cross-sectional* analyses, IFN therapy for COVID-19 has had limited success[58,59]. In fact, some studies have suggested that IFN therapy may be deleterious since excessive type I IFN could trigger hyperinflammation[5,9]. The latter view is supported by our *longitudinal analysis*, which indicates that early upregulation of ISGs is predictive of progression to more severe disease. In summary, our results demonstrate that it may be overly simplistic to assume a uniformly negative correlation between type I IFN signaling and COVID-19 severity. Rather, our data support a more nuanced, and temporally variable, relationship between IFN response and severity, which may explain the mixed results from clinical trials of IFN as a therapeutic.

We observed profound shifts in blood cell type proportions in our cohort (Fig. 5). Multiple T-cell subtypes were depleted among severe COVID-19 patients, consistently with the known lymphopenia phenotype of COVID-19[3,9]. Clonal expansion of T cells was inversely proportional to disease severity, indicating a potential causal role for dysfunctional adaptive immune response[3] (Fig. 1I). Similarly, we observed major shifts in the myeloid compartment. HLA-DR[hi]CD11c[hi] inflammatory monocytes were expanded in mild COVID-19, whereas severe and critical cases showed increased neutrophil precursors (suggestive of emergency myelopoiesis), non-functional mature neutrophils, and HLA-DR[lo] monocytes[18]. cDCs decreased with increasing severity, potentially indicating aberrant coupling of the early innate immune response to the subsequent adaptive response[60–63].

Intriguingly, T-cell states enriched in severe and critical samples showed high expression of suppressors of type I IFN signaling: *SOCS3, SOCS2, ITGA6, PIM1,* and *PRDM1*. This trend was observed in subpopulations of multiple T-cell subtypes, including CD4[+] Tcm, CD4[+] Tem, naive CD4[+], and naive CD8[+] (Fig. 6 and Supplementary Data 5), and also confirmed in multiple other datasets (*SOCS3*), in PBMCs as well as in airway samples. Upregulation of these genes could potentially contribute to the widely reported deficiency in type I IFN signaling in severe COVID-19[3,64–66], which could in turn promote SAR-CoV-2 replication[33]. Indeed, inhibiting SOCS3 in vitro, using shRNAs as well as the osteoporosis drug zoledronic acid, consistently reduced replication of WT and Delta SARS-CoV-2. Thus, modulation of these upstream factors may represent a novel therapeutic strategy for COVID-19. More broadly, since SOCS3 has been identified as a virulence factor even in severe influenza, it is conceivable that these findings are also relevant to other viral diseases[67,68].

In summary, we have generated a unique longitudinal single-cell data resource across the entire course of active COVID-19, and uncovered surprising differences between the early- and late-stage implications of type I IFN response. Our results suggest that type I IFN signaling may have both protective and deleterious effects in COVID-19, depending upon timing and disease severity[69]. We have also shown that the IFN suppressor *SOCS3* is upregulated in blood and airway in severe COVID-19, and that its inhibition substantially attenuates SARS-CoV-2 replication in vitro. We anticipate that our molecular findings may lead to novel assays for COVID-19 patient monitoring and

stratification, and also potentially support the development of a host-directed therapeutic.

## Methods

### Patients and clinical sample collection
Clinical samples are obtained under the PROTECT protocol approved by the Singapore National Healthcare Group Domain-Specific Review Board (2012/00917). All participants provided written informed consent for sample collection and subsequent analyses. All relevant ethical regulations were closely complied with.

### Isolation of PBMC
The patient's blood samples were handled in accordance with Biosafety Level 3 safety requirements following risk assessments approved by Institutional Biosafety Committees. Approximately 8 mL of patient blood was collected in BD Vacutainer® CPT™ Mononuclear Cell Preparation Tubes-Sodium Heparin (BD, Cat no. 362753) and was centrifuged at 20 °C for 15 min at 530 x $g$. The plasma layer was aspirated and stored separately. The PBMC layer was transferred to a new tube with 9 mL of PBS and spun down at 300 RCF for 15 min. Following two washes with wash buffer (1% FBS, 1 mM EDTA), the resulting cell pellet was frozen down in freezing media (90% FBS, 10% DMSO), and stored at −80 °C.

### Thawing and resuspension of PBMC
Rapid thawing of frozen PBMC vials was conducted at 37 °C for 2 min on a heat block. PBMC were thawed with pre-warmed thawing media (RPMI + 5% Human Serum + 1% Penicillin/streptomycin + 1% glutamine) until the final volume was 10 mL and centrifuged at 300 × $g$ for 5 min at 4 °C. PBMC pellets were gently re-suspended in 5 mL of pre-warmed wash media (RPMI + 10% FBS + 1% Penicillin/streptomycin + 1% glutamine) and centrifuged at 200 × $g$ for 5 min at 4 °C. Following two washes with 3 mL of pre-warmed PBS + 0.04% BSA, cells were strained using a 100 μm Macs SmartStrainer (Miltenyi Biotec, Cat no. 130-110-917) to filter off cellular clumps and debris. Using a Scepter 2.0 Cell Counter (Life Science Research), cell concentration was determined. The cells from each sample were then centrifuged and re-suspended at a final concentration of $1 \times 10^6$/mL for use in sample batching. Each pool consisted of 16 different COVID-19 patients of varying demographics like age, sex, ethnicity, and varying COVID-19 severity. 100 μL of cell suspension from each of the 16 samples were pooled together to make a single pooled cell suspension ($16 \times 100 = 1600$ μL) for downstream single-cell RNA-sequencing, single-cell TCR, and BCR V(D)J sequencing. (Protocol is available at https://www.protocols.io/view/demuxlet-cell-preparation-protocol-bf87jrzn).

### scRNA-seq, TCR and BCR V(D)J sequencing
Single-cell capturing and downstream library construction were performed using Chromium Single Cell 5′ Version 2 Reagent kits (10X Genomics). 70,000 cells from a single pool of 16 COVID-19 patients were each loaded into 2 wells of the Chromium Next GEM Chip K Single Cell Kit (10X Genomics, Cat no. 1000286) as technical replicates and subsequently into the Chromium Single Cell Controller (10X Genomics) for partitioning of single cells into Gel Bead-In-Emulsions (GEMs). The polyadenylated mRNA was then reverse transcribed (RT) into 10x barcoded full-length cDNA containing unique cell

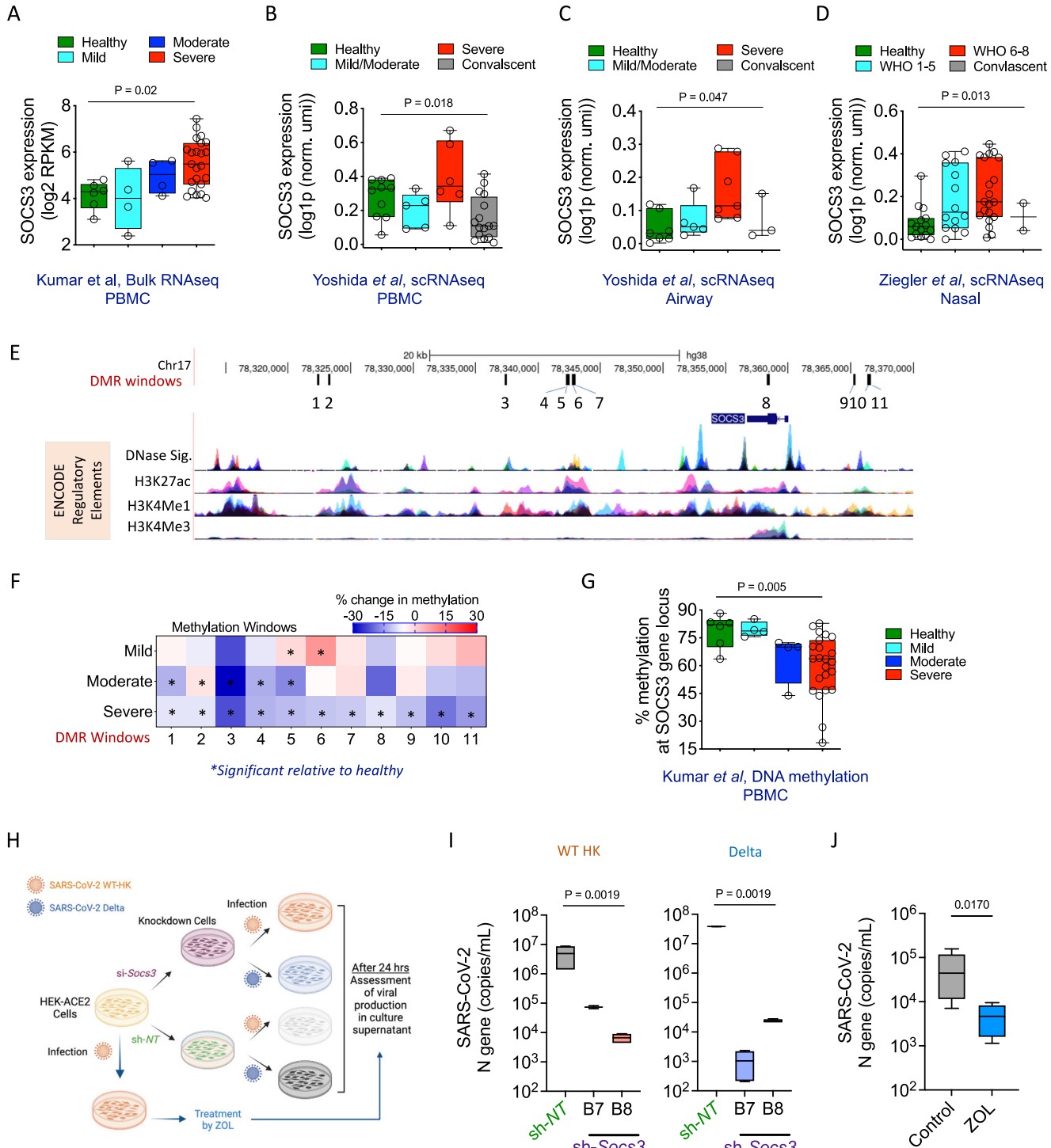

**Fig. 7 | SOCS3 expression associates with severe COVID-19 and enhances SARS-CoV-2 replication in vitro. A** *SOCS3* expression in healthy, mild, moderate, and severe COVID-19 donors, inferred from bulk RNA-seq analysis of PBMCs (*N* = 37, Singapore cohort). **B** Pseudobulk *SOCS3* expression in healthy, mild/moderate, severe, and convalescent COVID-19 donors, inferred from PBMC scRNA-seq by averaging across cells[4] (*N* = 37). **C** Upper airway[4] (*N* = 22). **D** Nasal swab, healthy, WHO COVID-19 severity 1–5, severity 6–8, convalescent, similar to ref. 5 (*N* = 51). **E** *SOCS3* locus: 11 differentially methylated regions (DMRs) between PBMCs from severe COVID-19 and healthy donors, inferred from whole-genome bisulfite sequencing (black ticks). ENCODE chromatin profiles from 7 cell lines are shown below. **F** Degree of differential methylation at the 11 loci in mild, moderate, and severe COVID-19, relative to healthy. *: FDR *Q*-value ≤ 0.05; Kruskal–Wallis test.

**G** Similar to **A**: average of methylation levels at the 11 DMRs in the 37 donors. **H** Schematic of in vitro knockdown assay to evaluate the modulation of replication of two strains of SARS-CoV-2 by SOCS3: WT-HK and Delta. **I** Abundance of SARS-CoV-2 (qRT-PCR) in the cell culture supernatant after shRNA knockdown, with non-targeting shRNA (sh-NT) as a control. Two independent experiments were performed on two distinct clones of the HEK-ACE2 cell line (B7, B8). Data from one experiment is shown. **J**. Effect of small-molecule inhibition of SOCS3 with zoledronic acid (ZOL), similar to **I**. Error bars represent SE of 4 technical replicates. *p*-values in **A**–**D**, **G** and **I**: Kruskal–Wallis test. *p*-value in **J**: Student's *t*-test (two-sided). Box-and-whisker plots in this figure show the median (center line), 25th, and 75th percentile (lower and upper boundary), with a 1.5x inter-quartile range indicated by whiskers and outliers shown as individual data points.

barcodes and unique molecular identifiers. All of the above steps were performed in a BSL3 facility. The generated cDNA, which was heat-inactivated with an additional treatment at 60 °C for 30 min during the GEM-RT step, was transported to the Biosafety Level 2+ (BSL2+) laboratory for downstream cDNA amplification and library construction. Using Chromium Single Cell 5' reagent kit v.2 (10X Genomics, Cat no. 1000263) and Dual Index Kit TT Set A (10X Genomics, Cat no. 1000215), 5' gene expression libraries were constructed following the manufacturer's protocols.

Full-length TCR V(D)J and BCR V(D)J segments were enriched from amplified cDNA using Chromium Single Cell Human TCR Amplification Kit (10X Genomics, Cat no. 1000252) and Single Cell Human BCR Amplification Kit (10x Genomics, Cat no. 1000253) respectively following manufacturer recommendation. Subsequently, gene expression libraries, TCR, and BCR libraries were quantified using a 2100 Bioanalyzer with High Sensitivity DNA kit (Agilent Technologies, Cat no. 5067-4626). All the constructed libraries were sequenced in a paired-end mode (Read 1: 26 cycles, i7 index: 10 cycles, i5 index: 10 cycles, Read 2: 90 cycles) with an S4 flow cell using Novaseq 6000 sequencer on the Illumina platform.

### Genomic DNA isolation for genotyping
After single-cell sequencing, genomic DNA was extracted from the leftover cells from individual patients (not the pooled cell suspension) using QIAamp DNA Mini Kit (Qiagen, Cat no. 51306) following the manufacturer's instructions. Genomic DNA was eluted in 100 uL of RNase-, DNase-free water. Genotyping was performed on Illumina Global Screening Array-48 + v3.0 Bead Chip (Illumina, Cat no. 20030773) according to the manufacturer's protocol.

### Genotyping data analyses
Genotypes were called and saved into plink files (PED and MAP formats) from raw intensity data files using GeomeStudio (version 2.0). Subsequently, strand correction was performed using StrandScript to convert all Illumina genotyping records into the reference forward strand[70]. The strand-flipped plink files were converted into VCF files using plink v1.9[71]. Then genotype imputation was conducted using Beagle v5.1[72] coupled with East and south Asian SNPs from 1000 Genomes.

### scRNA-seq computational quality check (QC) analyses
Sequence reads in the raw fastq file were aligned to the human genome reference hg38 using CellRanger (version 5.0.0). As a first QC step, barcodes with a number of detected genes (NODG) less than 300 were filtered out. Afterwards, by taking advantage of genotyping data, Demuxlet[28] was utilized to identify inter-sample doublets in each library. Intra-sample heterotypic cell-type doublets were additionally identified and eliminated using DoubletFinder[73] from each library. After doublet removal, the remaining singlets across all libraries were clustered using the reference component analysis v2 (RCA2)[29,30] with graph-based clustering (Resolution 2, Supplementary Fig. 28). Expression levels of well-documented marker genes in PBMC major cell types were profiled across all clusters. Accordingly, cell clusters lacking expression for any major immune cell mark genes were removed. These removed clusters also showed low NODG profiles (Supplementary Fig. 28D), suggesting they were low-quality cells. After removing these clusters and then re-clustering, we further discarded a cluster (467 cells, Supplementary Fig. 29) with high expression levels of both B-cell and Platelet marker genes, i.e., *MS4A1* and *ITGA2B*, respectively, as well as a cluster of red blood cells (117 cells, Supplementary Fig. 29). Eventually, the PBMC major cell types were annotated using RCA2, namely NK cells, T cells, B cells, monocytes, conventional dendritic cells (cDC), plasma B cells, and platelets. Subsequently, a second round of QC was conducted based on NODG and the percentage of the mitochondrial gene (pMito) for each major cell type (Supplementary Fig. 30). Furthermore, we discarded samples with post-QC cells lower than 200.

### NK and T-cell sub-clustering analyses
Integration of major NK and T cells characterized by the RCA2 protocol across 20 batches was conducted using the reference-based reciprocal principal component analysis (PCA) implemented in Seurat v3.2.3[74]. The batch with the highest cell number was used as a reference for integration. Integrated data was clustered in an unsupervised manner with graph-based clustering (Resolution 2.5). Clusters with substantially low NODG (median NODG < 1000) were discarded (Supplementary Fig. 31). Subsequently, expression levels of putative marker genes in NK and T cells were profiled across all clusters (Supplementary Fig. 32). Based on the marker gene expression profiles, we sub-clustered CD4+ T cells into 5 sub-cell types (Supplementary Fig. 32): (1) Naive CD4+ T cells, which highly expressed *CCR7*, *TCF7*, *LEF1*, and *SELL*[10,75,76]; (2) central memory CD4+ T cells (CD4 Tcm), expressing high levels of *CCR7* as in Naive CD4+ T, but also a high expression of *CD69* compared to the latter[18]; (3) effector memory CD4+ T cells (CD4 Tem), which had high expression levels of *CCR6*, *CXCR6*, *CCL5*, *PRDM1*, and *S100A4*[18,75]; 4) regulatory T cells (Treg) that highly expressed *FOXP3* and *IL2RA*[75]; and 5) cytotoxic CD4+ T cells, characterized by high levels of *GZMB*, *PRF1*, and *GNLY*[40]. Likewise, CD8+ T cells were subdivided into 3 groups: (1) Naive CD8+ T cells, which highly expressed *CCR7*, *TCF7*, *LEF1*, and *SELL*[10,75,76]; (2) effector memory CD8+ T cells, characterized by high expression of *GZMK*[18]; and (3) cytotoxic CD8+ T cells with high expression levels of *GZMB*, *PRF1*, and *GNLY*[10]. Finally, NK cells, highly expressing *NCAM1* (CD56)[10], were segregated into two groups based on relative expression levels of *FCGR3A* (CD16) - CD16 bright and dim NK cells (Supplementary Fig. 32).

### Monocyte and cDC sub-clustering analyses
Integration of major monocytes and cDCs characterized by the RCA2 protocol across 20 batches was conducted using the reference-based reciprocal PCA implemented in Seurat v3.2.3[74] (Supplementary Fig. 33). The batch with the highest cell number was chosen as a reference for integration. Integrated data was clustered in an unsupervised manner with graph-based clustering (Resolution 1.5). Clusters with substantially low NODG (median NODG < 1000) were discarded (Supplementary Fig. 33). Monocytes were sub-clustered into 3 groups based on expression levels of CD14 and CD16 (Supplementary Fig. 34): 1) CD14+ monocytes (classical monocytes), with high expression of CD14 but low levels of CD16; 2) intermediate monocytes, which expressed both CD14 and CD16; and 3) CD16+ monocytes (non-classical monocytes), with high expression of CD16 but low levels of CD14[77]. cDC was characterized using the marker gene *CD1C*[78] (Supplementary Fig. 34).

### B-cell sub-clustering analyses
Integration of major B cells characterized by the RCA2 protocol across 20 batches was conducted using the reference-based reciprocal PCA implemented in Seurat v3.2.3[74] (Supplementary Fig. 35). The batch with the highest cell number was chosen as a reference for integration. Integrated data were clustered in an unsupervised manner with graph-based clustering (Resolution 1). Clusters with substantially low NODG (median NODG < 1000) were discarded (Supplementary Fig. 35). Afterward, B cells were subdivided into 2 groups (Supplementary Fig. 36): (1) naive B cells with high expression levels of *IL4R*, and (2) memory B cells, which highly expressed *CD27*[18].

After all QC steps, the median number of high-quality PBMCs per sample was 1105 (Supplementary Fig. 37).

### Differential cell-state enrichment analyses across various clinical severities
In order to identify cell state enrichments that were associated with clinical severity, each cell was assigned a value to measure the enrichment fold-change between each severity condition and a reference condition surrounding this cell. Here, we used Mild1 as a

reference instead of Asymptomatic, due to the low sample number in the latter group. In our study, we have 4 temporal stages for each severity (except Asymptomatic), namely days 1–4, days 5–8, days 9–14, and days >14. To avoid confounding by temporal stage (disease duration), we analyzed a subset of samples in each severity category. Subsets were randomly selected in such a manner as to ensure that the distribution of disease durations was matched across severity categories (Supplementary Data 7). Downsampling was conducted 100 times and the average enrichment fold-change was calculated to represent the differential cell state enrichments between two severities. Precisely, the enrichment fold-change was calculated using the following formula:

$$E_i = \log_2\left(\frac{\frac{N_i+1}{N_{mild1}+1}}{\frac{C_i}{C_{mild1}}}\right) \qquad (1)$$

$E_i$ is enrichment fold-change for severity $i$, and $i$ can be Asymptomatic, Mild2, Moderate, Severe, or Critical conditions. $N_i$ denotes the number of cells as severity $i$ in the neighboring 300 cells and $C_i$ indicates the total number of cells as severity $i$. Specifically, for each cell, we obtained its neighboring 300 cells in gene expression space and calculated the numbers of cells as Asymptomatic, Mild1, Mild2, Moderate, Severe, or Critical. Subsequently, we compared neighboring cell numbers in each severity with the Mild1 reference (pseudo-count is 1). In the end, we adjusted this raw enrichment ratio by dividing it with the total cell number ratio between each severity and Mild1 reference and then performed a logarithmic transformation (base is 2). Based on the above-mentioned strategy, we calculated average enrichment fold-changes after performing 100 times downsampling across sample temporal stages. Then cell states that were enriched and depleted in advanced severity were further identified. To achieve it, firstly, we selected cell states whose enrichment fold-changes were greater than 2 (in log2-transformed) in at least one of the comparison conditions (Mild2 vs Mild1, Moderate vs Mild1, Severe vs Mild1, and Critical vs Mild1), and clustered these cell states based on these varying enrichment fold-changes across these 4 comparisons using Mfuzz[79]. Secondly, for each obtained cluster, we took a mean of logarithmic enrichment fold-change across all cell states for each severity comparison and calculated Pearson correlation across these mean fold-changes (adding 0 as the first data point representing Mild1 reference). If the Pearson correlation is greater than 0.5, cell states in the cluster are considered enriched in advanced severity; if the Pearson correlation is less than -0.5, cell states in the cluster are considered depleted in advanced severity. For each cell subtype, marker genes in the cell states enriched/depleted in advanced severity were characterized using RCA2's DEG calling function between these cell states and the remaining cell states with the following parameters: min.pct=0.1, logfc.threshold=0.25 and p_val_adj≤0.1. Subsequently, we performed gene ontology (GO) term analysis in each cell subtype using EnrichR implemented in GSEApy[80]. Here we used genes expressing at least 10% of cells in each cell subtype as background gene list for GO analysis.

### Differential gene signature analyses between Progressor and Non-Progressors

DEGs between cells from Progressors (Mild1 and Mild2) and Non-Progressors were identified using single-cell DEG calling function in the RCA2 package with the following parameters: min.pct=0.1, logfc.threshold=0.25 and p_val_adj≤0.1 (Supplementary Data 3). DEG calling was performed for each cell type. In the end, we took the union of up/down-regulated gene lists across all cell types and then performed GO enrichment analysis using EnrichR implemented in GSEApy[80]. Here, we used genes that were expressed in at least 10% of cells as background for GO analysis.

### Differential gene signature analyses between baseline and next stage of Progressors

DEGs between cells from baseline and the next stage of disease Progressors (Mild1 and Mild2) were identified using single-cell DEG calling function in the RCA2 package with the following parameters: min.pct=0.1, logfc.threshold=0.25 and p_val_adj≤0.1. DEG calling was performed for each cell type. In the end, we took the union of up/down-regulated gene lists across all cell types (Supplementary Data 3) and then performed GO enrichment analysis using EnrichR EnrichR implemented in GSEApy[80]. Here, we used genes that were expressed in at least 10% of cells as background for GO analysis.

### Differential gene signature analyses for disease Non-Progressors across different time points

To identify gene signatures of temporal resolution for each COVID-19 severity, we characterized cell states that were enriched/depleted across different temporal stages. For each severity, Non-Progressor, each cell was assigned with a value to measure the enrichment fold-change between each temporal stage and reference stage surrounding it. Here, we chose samples collected in days 1–4 (after symptom onset) as a reference stage for Mild1, Mild2, and Moderate, but days 5–8 for Severe and Critical due to only one sample in days 1–4. Similar to formula 1, in each cell, we obtained its neighboring 300 cells in gene expression space and counted the numbers of cells across 4 temporal stages for each severity. Subsequently, we calculated the fold-change of neighboring cell numbers between each temporal stage and reference stage (pseudo-count is 1). In the end, we adjusted this raw enrichment fold-change by dividing it with the total cell number ratio between each stage and reference stage and then performed a logarithmic transformation (base is 2). Then cell states enriched and depleted across time were further identified. To achieve it, firstly, we selected cell states whose enrichment fold-changes were greater than 2 in at least one of the temporal stage comparisons and clustered these cell states based on these varying enrichment fold-changes using Mfuzz[79]. Secondly, for each obtained cluster, we took the mean of logarithmic enrichment fold-change across all cell states for each timepoint comparison and calculated Pearson correlation across these mean fold-changes (adding 0 as the first reference stage). If the Pearson correlation is greater than 0.5, cell states in the cluster were considered enriched across time; if the Pearson correlation is less than -0.5, cell states in the cluster were considered depleted across time. For each cell subtype, marker genes in the cell states enriched/depleted across time were characterized using RCA2's DEG calling function between these cell states and the remaining cell states with the following parameters: min.pct=0.1, logfc.threshold=0.25 and p_val_adj≤0.1. Subsequently, we took the union of marker genes across all cell subtypes (Supplementary Data 3) and performed GO analysis using EnrichR implemented in GSEApy[80]. Here we used genes expressing at least 10% of cells in each cell type as background gene list for GO analysis.

### Metagene analysis of 13 prognostic type I IFN genes

For each gene, we calculated the mean of normalized expression value across all cells in a given sample and then computed $z$-scores of these mean values across all samples. In the end, for each sample, we took a 10% trimmed mean of $z$-scores across all type IFN genes to represent its type I IFN metagene expression level. In particular, we performed $z$-transformation across samples in the same disease duration and severity, i.e., Mild1 day 1–4, Mild1 day 5–8, and Mild2 day 5–8.

### Prognostic marker validation analysis in a German cohort

PBMC scRNA-seq data from an independent cohort[18] were downloaded via the Fastgenomics portal (https://www.fastgenomics.org/news/fg-covid-19-cell/). We identified two Mild Progressor (626 and 1751 cells) and two Mild non-Progressor samples (3465 and 4420 cells)

from Days 1–8 post-symptom onset. In each sample, we averaged the expression values of each gene across cells (pseudobulk). Then we zero-centered and scaled the average expression values for each gene across all samples so that its mean across all samples was zero and variance was one (z-score). Finally, we calculated the prognostic IFN metagene score of each sample as the 10% trimmed mean of the pseudobulk z-scores of the 13 prognostic marker genes.

## Multiplex microbead-based immunoassay

Across the 83 samples, we correlated the mRNA-based prognostic IFN metagene score (13 genes) with plasma expression levels of cytokines and chemokines. Plasma samples were treated with 1% Triton X-100 solvent–detergent mix for virus inactivation and immune mediators levels were measured with the Luminex assay using the 45-plex Human ProcartaPlex Panel 1 (ThermoFisher Scientific; USA), as described earlier by us[26]. We then tested the three highly correlated plasma markers (IFN-alpha, MCP-1, IP-10) for prognostic efficacy by comparing their expression levels in Progressors and non-Progressors. We excluded one Mild2 progressor who was sampled during Days 1–4 since there were no additional progressors of the same type (Fig. 1C).

## Cell abundance analyses

Samples were separated into two groups based on collection date after symptom onset, namely Day 1–8 and >Day 8. For each major cell type as well as sub-cell type in NK, T, B, and Myeloid cells, we compared cell abundance across samples from each disease severity category.

## TCR and BCR repertoire analyses

TCR and BCR V(D)J datasets were analyzed using CellRanger vdj function (version 5.0.0). Based on the generated filtered contig annotation files, TCR/BCR clonotype was determined for each T/B-cell, respectively. Specifically, to define a valid TCR, both alpha (TRA) and beta (TRB) chains should be detected in a T-cell; likewise, a valid BCR was determined within a B-cell if both heavy and light chains were detected. Due to the distinct regulation of allelic exclusion at the transcription level between alpha and beta chains[81], we maintained up to two alpha chains and only one beta chain with top UMI counts to define a TCR clonotype. Similarly, only one heavy and light chain with top UMI counts was retained for a B-cell due to the allelic exclusion mechanism[82]. Based on the defined TCRs/BCRs in each sample, their clonality expansion proportions were computed by dividing the frequency of each TCR/BCR by the total number of detected TCRs/BCRs. Here, we only considered samples with detected TCRs no less than 50 for TCR analysis and with detected BCRs no less than 30 for BCR analysis. To compare the TCR clonality index across different COVID-19 clinical severities, we selected the top 50 unique TCR clonotypes ranked by frequency in each sample and calculated the proportion of the top 5 TCRs among them. Similarly, we selected the top 30 unique BCR clonotypes ranked by frequency in each sample and calculated the proportion of the top 5 BCRs among them.

## DNA and RNA isolation from TRIzol

Total RNA and DNA were extracted from PBMCs lysed in TRIzol (Thermo Fisher Scientific, Cat no. 15596026) by the acid guanidinium thiocyanate-phenol-chloroform extraction method. This was performed on 37 individuals from an independent Singapore cohort spanning healthy controls, mild, moderate, and severe COVID-19 (in this cohort, severe and critical samples were combined into a single category). The aqueous phase was processed using the Qiagen RNeasy Micro clean-up procedure to isolate RNA. The leftover interphase and organic phase were further processed using the Back extraction protocol (Thermo Fisher Scientific) to isolate genomic DNA. RNA and genomic DNA concentrations were estimated using PicoGreen (Cat no. P7589).

## Singapore cohort bulk RNA sequencing and data analysis

For bulk RNA sequencing of PBMC samples from the independent Singapore cohort (see above), we performed reverse transcription and amplification using a standard protocol[83]. Briefly, we synthesized cDNA from 2 ng of purified total RNA using modified oligo(dT) primers, and cDNAs were further amplified. The quantity and integrity of cDNA were assessed using the DNA High Sensitivity Reagent Kit (Perkin Elmer: LabChip GX). Subsequently, pooled cDNA libraries were prepared (250 pg of cDNA per sample, Illumina Nextera XT kit, Cat no. FC-131-1096) with dual indices for demultiplexing. The libraries were quantified using qPCR (Kapa Biosystems) to ascertain the loading concentration. Samples were subjected to an indexed PE sequencing run of 2 × 151 cycles on an Illumina HiSeq 4000. We mapped paired-end reads to human genome build GRCh38 using the STAR aligner[84] and counted reads mapped to genes using featureCounts[85] and GENCODE v31 gene annotations[86]. We quantified gene expression as log2-transformed reads per exonic kilobasepair per million mapped reads (log2RPKM) using the edgeR Bioconductor package[87]. Gene expression estimates for *SOCS3* were then analyzed for association with disease severity (D.K. et al.[49], unpublished observations).

## *SOCS3* expression in previously published scRNA-seq datasets

PBMC and airway single-cell RNA-seq data from COVID-19 samples[50,51] were obtained from the COVID-19 Cell Atlas (www.covid19cellatlas.org) and processed using Scanpy[88] to infer gene expression levels. *SOCS3* expression of each sample was calculated by averaging across all single cells. Statistical significance was calculated using the Kruskal–Wallis test (non-parametric ANOVA).

## Whole-genome bisulfite sequencing

Genomic DNA from the independent Singapore cohort (see above) was converted through sodium bisulfite treatment using the DNA Methylation-Direct Kit (Zymo Research, Cat no. D5021) according to the manufacturer's instructions. Converted genomic DNA was subjected to PCR-free library construction using PBAT (Post-Bisulfite Adaptor Tagging[89]) optimized for low-input and damaged genomic material (OptPBAT; D.K. et al.[49], unpublished observations). Library concentration was estimated using the KAPA library quantification kit (Kapa Biosystems, Cat no. 07960140001), and libraries were pooled at 2 nM for indexed PE sequencing with 2×151 cycles on an Illumina NovaSeq 6000 using custom primers: 5'-GTA AAA CGA CGG CCA GCA GGA AAC AGC TAT GAC-3' and 5'-GTC ATA GCT GTT TCC TGC TGG CCG TCG TTT TAC-3'.

## Differential methylation analysis

FASTQ-format reads from whole-genome bisulfite sequencing were first trimmed using Trim Galore with an additional 20 bp trimmed on either side. The trimmed reads were then aligned to the human genome GRCh38 build using Bismark in non-directional mode for each of the read pairs[90]. We then identified differentially methylated regions (DMRs) in mild, moderate, and severe samples relative to healthy controls using a 200 bp sliding window with a 100 bp step size[91]. DNA methylation was then quantified as the percentage of methylated CpG reads in each window using methylKit[92]. CpG sites were retained if at least one corresponding genomic window contained at least 10 reads. In each window, DMRs between sample groups were estimated using the function "calculateDiffMeth" in methylKit. Only widows where at least 50% of the samples had methylation estimates (at least 10 reads) were considered in the DMR analysis. Furthermore, we only tested windows for differential methylation if they were highly variable across samples. DMR p-values were adjusted for multiple testing[93] and then defined as significant at a Q-value threshold of 0.05 and at least a difference of 10 percentage points in mean CpG methylation between the two groups.

## Cell lines and virus

HEK293T cells expressing human ACE2 (HEK-ACE2, BEI Resources, Cat no. NR-52511) were cultured in complete media containing Dulbecco's modified Eagle medium (Cell Clone, Cat no. CC3004) with 10% CELLect FBS Gold (MP Biomedicals, Cat no. 2916754), 100 IU/mL penicillin, 100 μg/mL streptomycin and 0.25 μg/mL amphotericin-B (Sigma, Cat no. A5955). SARS-CoV-2 isolates (Cat no. NR-52282, Hong Kong/VM20001061/2020, WT; Cat no. NR-55671, hCoV-19/USA/MD-HP05285/2021, Delta variant), were obtained from BEI Resources and propagated and titrated using the standard plaque assay in Vero E6 cells. All experiments on live SARS-CoV-2 virus were performed at Biosafety Level 3.

## Effect of SOCS3 inhibition on SARS-CoV-2 replication

For *SOCS3* knockdown, we used gene-specific shRNAs from the RNAi Consortium (TRC) library (Sigma-Aldrich, USA). HEK-ACE2 cells were transfected in a 24-well plate with 500 ng of shSOCS3#7 (5′-CCG GCG GCT TCT ACT GGA GCG CAG TCT CGA GAC TGC GCT CCA GTA GAA GCC GTT TTT G-3′), shSOCS3#8 (5′- CCG GCT CCT ATG AGA AAG TCA CCC ACT CGA GTG GGT GAC TTT CTC ATA GGA GTTT TTG-3′) or non-targeting control (ShNT), diluted in serum and antibiotic-free Opti-MEM (Gibco, Cat no. 51985034) using Lipofectamine 3000 reagent (Invitrogen, Cat no. L3000-015) as per the manufacturer's protocol. Knockdown efficiency was quantified using qRT-PCR (SOCS3-FP: 5′-CAA GGA CGG AGA CTT CGA TT-3′, SOCS3-RP: 5′- AAC TTG CTG TGG GTG ACC AT-3′; B2M-FP: 5′- GCCCAAGATAGTTAAGTGGGATCG-3′, B2M-RP: 5′- TCA TCC AAT CCA AAT GCG GC-3′). Cells were incubated for 36 h before infection with SARS-CoV-2 at an MOI of 0.1 for 1 h. Virus inoculum was removed after 1 h and infection medium (DMEM with 2% FBS) was added to the cells and incubated further for 24 h. After 24 h, the supernatant was processed for viral RNA isolation (mdi, Cat no. VLRKXXXXXXX0100), and viral titer was quantified using qRT-PCR (SYBR green) for the SARS-CoV-2 N-gene (FP: 5′-CAC ATT GGC ACC CGC AAT C-3′, RP: 5′-GAG GAA CGA GAA GAG GCT TG-3′). Viral copy number was estimated by generating a standard curve using SARS-CoV-2 genomic RNA of known titer.

To examine the effect of treatment with zoledronic acid (ZOL), HEK-ACE2 cells were pretreated with 50 μM ZOL for 2 h and subsequently infected with SARS-CoV-2 (Hong Kong isolate). After 48 h, the supernatant was processed for an estimated viral copy number as above.

## Reporting summary

Further information on research design is available in the Nature Portfolio Reporting Summary linked to this article.

## Data availability

Raw sequencing data generated in this study are available via the European Genome-Phenome Archive (EGA): Study—EGAS00001005545; Dataset—EGAD00001007995. Processed data are available via the *Zenodo* (https://doi.org/10.5281/zenodo.5153528). All other data are available in the article and its Supplementary files or from the corresponding author upon request. Source data are provided in this paper.

## Code availability

Custom codes used in the study are available via the GitHub (https://github.com/prabhakarlab/SCAN_COVID19).

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

## Acknowledgements

We are grateful to Nirmala Arul Rayan for initial guidance on single-cell and PBMC protocols and Shvetha Sankaran, Swaine Chen, and Muhammad Aminullah for setting up the BSL2+ lab, as well as Khor Chiea Chuen for assistance in genotyping. This study was supported by the following grants: CDAP201703-172-76-00056, IAF-PP-H18/01/a0/020, and ACCL/19-GAP064-R20H-H from the Agency for Science, Technology, and Research (A*STAR), Singapore; COVID19RF-001, COVID19RF-003, COVID19RF-060 and OFLCG19May-0034 from the Singapore National Medical Research Council COVID-19 Research Fund, NRF2016NRF-NSFC002-013, NRF2018NRF-NSFC003SB-002 from the Singapore National Research Foundation, and IA/S/16/2/502700 from the Wellcome Trust/DBT India Alliance.

## Author contributions

S.P., A.S., and K.R. conceived the study. S.P., A.S., D.L., D.R., Q.X.X.L., B.Y., L.F.W., and K.R. designed the study. A.M.G., D.R., and A.S. designed the biosafety protocols. D.R., A.M.G., L.M.T., P.N.V., and W.O.Y.C. generated single-cell data. S.W.F., L.R., and L.N. generated Luminex data. Q.X.X.L. developed the computational pipelines. Q.X.X.L. and D.R. analyzed the data, with support from L.M.T. B.Y., D.L., K.R., J.S., L.C., L.J.S., and S.Y.T. constructed the age- and sex-matched cohort used in this study, provided samples, and guided clinical metadata analysis. R.A. and A.S. performed in vitro infection experiments; D.K., Y.C., and B.L. generated bulk RNA-seq and DNA methylation data; D.R., Q.X.X.L., S.P., and A.S. wrote the manuscript. All authors contributed to the editing of the manuscript.

## Competing interests

The authors declare no competing interests.

## Additional information

[1]Laboratory of Systems Biology and Data Analytics, Genome Institute of Singapore, Agency for Science, Technology and Research (A*STAR), Singapore 138672, Singapore. [2]Programme in Emerging Infectious Diseases, Duke-NUS Medical School, Singapore 169857, Singapore. [3]Singapore Immunology Network, A*STAR, Singapore 138648, Singapore. [4]Department of Microbiology and Cell Biology, Centre for Infectious Disease Research, Indian Institute of Science, Bangalore 560012, India. [5]A*STAR Infectious Diseases Labs (A*STAR ID Labs), A*STAR, Singapore 138648, Singapore. [6]Alexandra Hospital, Singapore 159964, Singapore. [7]Changi General Hospital, Singapore 529889, Singapore. [8]Division of Infectious Diseases, Department of Medicine, National University Health System, Singapore 119228, Singapore. [9]Yong Loo Lin School of Medicine, National University of Singapore, Singapore 117597, Singapore. [10]Lee Kong Chian School of Medicine, Nanyang Technological University, Singapore 636921, Singapore. [11]National University Hospital, Singapore 119074, Singapore. [12]SingHealth Duke-NUS Global Health Institute, Singapore 168753, Singapore. [13]National Centre for Infectious diseases, Singapore 308442, Singapore. [14]Tan Tock Seng Hospital, Singapore 308433, Singapore. [15]These authors contributed equally: Quy Xiao Xuan Lin, Deepa Rajagopalan. [16]These authors jointly supervised this work: Amit Singhal, Shyam Prabhakar. ✉e-mail: amit_singhal@idlabs.a-star.edu.sg; prabhakars@gis.a-star.edu.sg

