## [Peer Review File · Nature Communications]

Longitudinal Single Cell Atlas Identifies Complex Temporal Relationship between Type I Interferon Response and COVID-19 SeverityREVIEWER COMMENTS

Reviewer #1 (Remarks to the Author):

Lin et al., performed scRNA-seq to examine longitudinal changes in cell signalling and subset frequency in COVID-19. Upregulation of the type I IFN signalling pathway was a key factor that distinguished patients that progressed from those that did not. This finding is interesting and unexpected, since most available data suggests that type I IFN signalling is reduced in severe/critical relative to mild COVID-19 (as the authors also observe in their cross-sectional analysis). Upregulation of type I IFN signalling was transient and decreased over time in both progressors and non-progressors. IFN signalling was reduced in severe/critical relative to mild COVID-19 and the authors suggest SOCS3 upregulation as a possible mechanism for this reduction.

Overall, the study was well-performed and the statistical analysis is appropriate, and it is valuable to have longitudinal samples collected from early in disease. While the data and findings are of interest, I have some reservations as to how generalisable the data is (due to the limited number of patients that progressed and their heterogeneous disease courses) and how useful type I IFN upregulation will be clinically as a biomarker for progression to severe disease. I am also not convinced there is enough evidence to indicate that SOCS3 is functioning to suppress type I IFN signalling in COVID-19. I discuss these issues in more detail below.

Main comments:

For the longitudinal analyses, the authors have focussed on patients with mild disease (Mild1 and Mild2) due to the availability of samples. However, most of these patients only progressed from Mild1 to Mild2 or from Mild2 to Moderate disease based on Figure 1C. Therefore I am not sure how useful early type I IFN upregulation is as a biomarker of patients that will progress to severe disease (which is ultimately what would be most useful clinically). Was type I IFN signalling also enriched in moderate/severe patients that progressed relative to those that didn't? Is there any association between early type I IFN signalling and risk of death/risk of long-term effects of disease? The timing for measuring type I IFN/type I IFN gene signatures would also be difficult on a practical level since patients

progress at different speeds and would need to be sampled before reaching their maximal disease severity.

While the overall number of patients in this study is large, the number of mild individuals that progressed to more severe disease is relatively small and the validation cohort only includes two progressors and two non-progressors. Therefore, while the findings from this study are interesting, further validation will be required to confirm that the observed result is generalisable.

Were all of the patients in this study infected with the same variant of SARS-CoV-2 and is this information available? This is a key factor that could affect the time course of the disease.

Is SOCS3 expression enriched overall in T cells and NK cells in severe/critical COVID-19 relative to mild COVID-19? Since DE analysis has been performed in e.g. subpopulations of T cell subsets, it is hard to grasp the overall importance of this finding. In which cell subsets is SOCS3 enriched in the scRNA-seq datasets analysed in Figure 7A-D?

The authors provide no direct evidence that SOCS3 is regulating type I IFN signalling. SOCS3 typically regulates signalling downstream of cytokines using the gp130 receptor, for example IL-6 and OSM. Do the authors have evidence for a non-canonical role of SOCS3 in COVID-19?

The author's message about the role of SOCS3 is a bit confusing. In Figure 6, SOCS3 is specifically upregulated in subsets T cells and NK cells, while in Figure 7, the authors assess its role in epithelial cells. The mechanism described in epithelial cells may be independent of type I IFNs since they were not measured in this assay. The authors should clarify their line of thinking and validate the inhibition of type I IFN signalling by SOCS3.

There are some relatively abundant T cell subsets missing from Figure 1G that are usually easily identifiable by scRNA-seq of PBMCs. These include MAIT cells and $V\delta 2 \gamma\delta$ T cells. Are these present within the dataset and if so, where are they located on the UMAP?

Were all T cells included in the clonality analysis performed in Figure 1? Since there are expected differences in clonality between different T cell subsets and the frequency of T cell subsets changes with disease, this analysis should be performed at the level of individual T cell subsets. Alternatively, the authors should show that their analysis is not influenced by the change in proportion of T cell subsets. The same should be done for the B cell analysis.

Minor comments:

To what stage of disease did the mild patients in the validation cohort progress to? It would be useful to have data from mild patients that developed severe/critical disease.

Figure 1D is misleading since it suggests that the Mild2 patients all progressed to critical disease, while this is only true for one patient based on Figure 1C.

Figure 1G – have these p-values been corrected for multiple testing? Do any of the other 41 plasma protein markers differ between progressors and non-progressors? i.e. the protein data should be analysed in an unsupervised manner rather than initially selecting proteins that correlate with the expression of the IFN gene signature – this may reveal other useful biomarkers for distinguishing progressors vs. non-progressors.

Figure 4 – do the authors have Luminex data to match Figure 4 i.e. showing the reduction in IFN α over time in progressors and non-progressors? This would further validate the finding.

Figure 4G/J/I – how does the metagene analysis performed here differ from the scores in Figure 3B-D? Is it the same analysis, but just showing the results for individual patients and time points? If so, this should be clarified in the text, as it currently implies that a new analysis was performed.

Line 292 – the authors should rephrase this sentence since the prognostic gene signature would not always be useful during Days 1-8 since one of the patients progressed during the first four days and another during the 5–8-day period.

It would be useful if Supplementary Table 1 could indicate which patients were included in

which analyses, since the authors subset the data in different ways throughout the paper. A summary figure displaying this information would also be helpful to the reader.

Figure 5 – have the p-values here been corrected for multiple testing?

Line 334 – should this be “coloured” rather than “clustered”. Otherwise I am not clear how the authors have clustered the data.

The top gene is missing from Supplementary Figure 2A.

Reviewer #2 (Remarks to the Author):

General:

In this manuscript, Lin et al describe the dynamic change of the immune cell profiles in response to SARS-CoV-2. The authors conducted a longitudinal analysis, especially focusing on early stages of infection. A total of 286 blood samples was collected from 108 COVID-19 patients and subjected to the single-cell level gene expression analysis. The patterns of TCRs and BCRs were also analyzed. One of the most relevant issues is that the previous studies which has analyzed a large number of longitudinally collected blood samples has been relatively rare so far. In addition to the series of transcriptome analysis, other types of multiple omics analyses were also conducted by microarray-based genotyping analysis, Luminex assays for the level of inflammatory cytokines, bulk RNA-seq and whole-genome bisulfite sequencing. To validate the deduced hypothesis, the authors also utilized some external cohort datasets. As a result of the integrative analyses of the collected datasets, they found that upregulation of type I IFN-stimulated genes (ISGs) should be the early signature of the subsequent worsening of the patients. There, the ISG expression was dynamic and rapidly receding to the original levels. In contrast, in severe and critical patients, the ISG expressions were deficient and IFN suppressors, such as SOCS3, were upregulated. The in vitro assays also showed that SOCS3 inhibition reduced SARS-CoV-2 replication.

Overall, I fully appreciate that the original blood samples collected for this study should be

substantial and valuable. However, since the samples are initially “pooled” and de-multiplexed based on the genomic variations later for the single cell analysis, I have a remaining concern that the produced datasets may have been substantially affected, thus imposing a problem on a precise analysis of the dynamic changes of discrete cell types and the gene expression changes within. Even admitting the analyses may have been conducted in a properly age-/gender-/severity- matched manner, the deduced story about the IFN-SOCS axis remains still controversial.

Major Comments:

1. To my knowledge, most of the described contents, except for the issues as described above, is confirmatory to or as expected from the previous studies (such as <https://doi.org/10.1016/j.xcrm.2020.100166>). Nevertheless, I consider the presented results should have important implications regarding the IFN-SOCS3 axis. However, I would like to point out that the story is still controversially and should need further extensive analyses before reaching to a particularly conclusion.

For example;

1-1: How does the longitudinal expression change for the SOCS3 gene or other IFN suppressor genes look like in the same individuals in vivo? Also, the authors should further inspect the behaviors of key genes at the critical time points both at the gene expression levels and at the cytokine levels.

1-2: Before conducting such an analysis, I’m concerned that the correlation between the scRNA-seq gene expression levels and their product cytokine levels is generally poor. The authors should further examine possible reasons with this regard.

1-3: Reasons of different IFN-SOCS3 responses at the early stage are not fully characterized. Were those individuals originally vulnerable for the induction or are there any other reasons? Would not the data or further extensive analysis using the samples collected after the convalescence period, or totally healthy controls, give a clue?

2. Technically, potential concerns related to the analysis using the “pooled” samples should be fully removed.

2-1: Although I understand there are careful validation analyses in the previous paper (ref

25), further detailed validation should be needed for this particular dataset shown to guarantee the clear de-multiplexing of the “pooled” samples based on the genomic polymorphisms. It is desirable that the close relatives or patients of the same ethnic backgrounds are not assigned to the same “pool”, but the genotyping analysis seems to have been done after or independently from the single cell analysis. I also wonder whether the de-multiplexing power should be sufficient for all the cell types, depending on the genomic polymorphisms on the mRNAs expressed on the respective cell types. For example, how many SNPs were utilized for the respective cell types and how discrete each sample was separated?

2-2: When the total number of 346,680 cells was divided by a total number of 286 individuals (Line 158), the resulting ~1,200 cell per sample should be too sparse for a current standard single cell RNA seq. This limited number of cells should be particularly problematic when TCR or BCR repertoires are analyzed. For those cases, exact overlap of particular serotypes is not expected between different individuals, thus using as many cells as possible (currently approximately up to 10,000 cells) is desirable for the reasonable analysis. Also, careful evaluation should be needed to confirm the sequencing depth should be sufficient and the representation of the cells from different individuals should be reasonably equal at the sufficient level for this study.

2-3: I also have a remaining concern to what extent the batch effects may have been compromised by the “pooling” approach.

Because of these reasons, at least, for some representative samples, a careful validation analysis should be conducted using individually prepared libraries.

3. It seems to me that the produced genotype and methylation type datasets, which could deepen the current study, are not fully utilized. In addition, it is a pity that methylation analysis is for a different group of patients. In fact, much more information could be extracted if the methylation analyses should be directly conducted for the same individuals used for the single cell analysis.

Minor Comments:

4. Figure 1: Although the authors annotated T cells and NK cells into detailed cell types in Figure 1, hereafter, they compared gene expression by four major cell types only. The authors should comment on whether the profiles are not different between those finer sub-classes.

5. Figure 1B: It seems that the samples are biased to symptomatic patients (only five individuals are recruited for “asymptomatic” cases). At least recently, is not it so difficult to recruit asymptomatic patients at the asymptomatic stage?

6. Figure 1C: In fact, the patients recruited for the current study showed various pattern of symptom patterns. As indicated by the authors, the variable host immune response to SARS-CoV-2 should be the key. Some of those response may accompany global changes on cellular populations. I wonder whether those factors can be analyzed from “pooled” samples as well. Even though the single cell analysis could not be conducted for this large number of samples, there should be some surrogate way, such as FACS or CyTOF, to confirm the global populations.

7. To elucidate the T cell response, the authors mainly analyzed the top ten clonotypes. It is unlikely that the key T cell population should consist such a large population, particularly at the early response, the analytical strategy should be re-considered to separate TCR patterns related to severity and/or progression. It is desirable to also conduct the repertoire sequencing in bulk and compare the results with each other.

8. Figures 2 and 6 are not always informative, thus, are not necessary. At least, they could be presented in a far more concise manner.

9. The term used in the manuscript should be unified; for example, SARS-CoV-2 (line 71) and SARS-CoV2 (line 475 and line 477).

10. Fig 1C: In the main text, the authors describe the number of mild1 and mild2 progressors only. Judging from the Figure, there seems also progressed patients in moderate and severe groups. Are there any particular difference between them?.

11. Fig 1E: The authors grouped PBMCs into seven major groups. For dendric cell, they only show cDC. However, previous scRNA-seq studies, using PBMC for COVID-19 analysis, generally separate pDC and cDC or collectively annotated them as DC. Is there any specific reason to depict only cDC for the annotation?

12. Fig 2: Although authors annotated cells into subtypes in Fig 1G- 1I, they identified DEGs based on major three cell types, T/NK, B and Myeloid cells. Even though the result demonstrated that the type I interferon signaling pathway is the common feature in all cell types, this method doesn't seem to take advantage of scRNA-seq. On the other hand, in Fig 6, they identified severity associated gene ontologies based on the detailed cell type. I wonder if there are any reasons for changing the analytical axes for the respective analyses?

Reviewer #3 (Remarks to the Author):

The authors performed a longitudinal single cell RNA-seq on 286 blood samples from age- and gender-matched COVID-19 patients, including 73 with early samples. The data presented in this work are extremely informative as the longitudinal design allows evaluating the response profile in patients by dividing them on the basis of the different course of the disease. The main result of the work is the description of the trend of the type1-IFN response in relation to the disease worsening. Differently from the results published in several papers (cited by the authors in discussion), the data presented here demonstrate a high interferon response in patients who subsequently worsen and show as this response is associated with the concentration of two plasmatic mediators associated with severe disease (IP-1 and MCP-1). There are some points that need clarification.

1) As well described in the discussion, a large literature reports association between a strong Type-I IFNs response and a mild COVID-19 disease. The data presented here support the opposite view, at least in the early stages of the disease. Can the timing 1-8 days from the onset of symptoms represent a good definition of "early response"? This is a long time frame including innate and adaptive immune induction.

2) Figure 2: The authors showed an increased expression of ISG in progressor Mild 1 patients

both at days 1-4 and at days 4-8. In contrast, this association in Mild 2 patients was only true at days 4-8. How do they explain this difference?

3) The authors identify 13 genes associated with the IFN response that are higher in patients who will get worse than in those who will remain stable. The data is certainly interesting but should be validated on a cohort wider than that shown here (4 total patients, 2 progressor and 2 non progressor). Please, add data about a larger validating cohort.

4) They defined the average normalized expression of these 13 genes at baseline as a prognostic score; please, specify baseline, is it day 1-8?

5) In figure 4G the authors showed a comparison of the expression of the 13 genes between different clinical severities. Levels of these genes in "moderate and mild 2" patients were higher in early times (0-4 and 4-8 days) than in "severe" and "mild 1" patients. Indeed, the figure shows that severe / critical patients showed levels of those 13 genes similar to those observed in mild patients, making difficult the hypotheses of an association between the severity and expression of these genes at T 5-8. To address this problem, the authors suggest that this is the main limitation of cross-sectional studies, while a longitudinal approach overtime allows identifying a higher ISG expression with a high risk of worsening. This aspect represents the main limitation in the use of the set of 13 genes proposed as an early marker of worse prognosis in the clinical management of COVID-19 patients. How could this system be exported to clinical practice? How to define in a single patient whether the early ISG value is decreasing or is stable? The wide timing defined "early" (days 1-8) makes it even more complicated. Please, add a comment on this point.

6) In lines 205-207, the authors summarize that the 13 genes discriminate well the mild patients that will get worse from those that will not get worse. Why the authors did not show the analysis of these 13 genes even in moderates who become severe or critical. From figure 1C it is clear that these type of patients were also included in the study? The question is: is the early high production of type-1 IFNs able to predict a worsening only in patients starting with an initial mild disease?

7) The data about SOCS expression are very interesting and can explain the reduced IFN- α expression and function in severe clinical presentation. At what time point the expression of SOCS was increased? Is this timing compatible with the ISG reduction described in the longitudinal analysis? Why the author did not include SOCS expression in the longitudinal analysis? Can the author verify the possible contribution of SOCS expression in defining

progressor vs non progressor patients? Is the SOCS expression associated with a reduction of plasmatic IFN- α ?

8) Linea 99-102. Several cell populations are described in severe COVID-19 and some of them can have a prognostic value in the further worsening of the pathology. Among these, the authors forgot to include the myeloid derived suppressor cells (MDSC) that expand early after infection and can predict a fatal outcome (doi: 10.1038/s41419-020-03125-1).

REVIEWER COMMENTS

Reviewer #1 (Remarks to the Author):

Lin et al., performed scRNA-seq to examine longitudinal changes in cell signalling and subset frequency in COVID-19. Upregulation of the type I IFN signalling pathway was a key factor that distinguished patients that progressed from those that did not. This finding is interesting and unexpected, since most available data suggests that type I IFN signalling is reduced in severe/critical relative to mild COVID-19 (as the authors also observe in their cross-sectional analysis). Upregulation of type I IFN signalling was transient and decreased over time in both progressors and non-progressors. IFN signalling was reduced in severe/critical relative to mild COVID-19 and the authors suggest SOCS3 upregulation as a possible mechanism for this reduction.

Overall, the study was well-performed and the statistical analysis is appropriate, and it is valuable to have longitudinal samples collected from early in disease. While the data and findings are of interest, I have some reservations as to how generalisable the data is (due to the limited number of patients that progressed and their heterogeneous disease courses) and how useful type I IFN upregulation will be clinically as a biomarker for progression to severe disease. I am also not convinced there is enough evidence to indicate that SOCS3 is functioning to suppress type I IFN signalling in COVID-19. I discuss these issues in more detail below.

Main comments:

For the longitudinal analyses, the authors have focussed on patients with mild disease (Mild1 and Mild2) due to the availability of samples. However, most of these patients only progressed from Mild1 to Mild2 or from Mild2 to Moderate disease based on Figure 1C. Therefore I am not sure how useful early type I IFN upregulation is as a biomarker of patients that will progress to severe disease (which is ultimately what would be most useful clinically). Was type I IFN signalling also enriched in moderate/severe patients that progressed relative to those that didn't?

Response: In the original manuscript, we focused on progression from Mild1 and Mild2, since the cohort included a substantial number of Mild Progressors (11 individuals) and Mild Non-Progressors (52) within the relevant range of disease duration. In contrast, we had limited statistical power to differentiate Moderate Progressors (2 individuals within Days 1-8) from Moderate Non-Progressors (4 individuals). We show below the 13-gene IFN signature for these 6 individuals with moderate disease at presentation. The IFN signature is indeed highest in one of the two Moderate Progressors – this individual progressed to Severe disease subsequently. However, due to variability across patients and the small number of individuals (6), the difference between Moderate Progressors and Non-Progressors was not significant (p-value=0.255; **Reviewer Figure 1**, see below).

To further strengthen the validation of the 13-gene signature, we identified one additional Mild Non-Progressor in the German cohort, which we had overlooked earlier. We have now updated **Figure 3E** by including this additional sample (see below). This has improved the prognostic significance of the 13-gene signature in the validation cohort (p-value=0.012 in revised manuscript vs p-value=0.037 in the original version). Moreover, the number of individually significant genes in the validation cohort has now increased from 7/13 to 8/13 (**Revised Figure 3E**, see below).

Reviewer Figure 1: 13-gene IFN response signature in 2 Moderate Progressors vs. 4 Moderate Non-Progressors at baseline, Singapore cohort. Only baseline samples collected During Days 1-8 are included.

Revised Figure 3E: Heatmap shows the scaled pseudobulk expression levels (expression z-score) of the 13 prognostic ISGs in baseline samples from 2 Mild Progressors and 3 Mild non-Progressors in the German cohort. *: p-value \leq 0.05, one-sided Student's t-test for upregulation in Progressors relative to non-Progressors. p-value above heatmap: difference between Progressors and non-Progressors in trimmed mean of the 13 genes (Student's t-test, one-sided).

Is there any association between early type I IFN signalling and risk of death/risk of long-term effects of disease?

Response: This is indeed a relevant readout. We obtained access to long-term clinical phenotypes of 14 Non-Progressors and 6 Progressors from our cohort. At Day 180 (6 months, convalescent phase), only one of these 20 individuals had symptoms. Given the low incidence of long-COVID in this sub-cohort, it was not possible to examine correlations between baseline gene expression and long-term phenotypes. As an alternative strategy, we examined Day 180 plasma luminex data from these 20 subjects. VEGF-A, a marker of long-COVID (Patel et al., 2022), was elevated in Progressors relative to Non-Progressors (p-value=0.049; two-sided unpaired t-test). However, this effect was no longer significant after correction for multiple testing, perhaps due to the low proportion of individuals with long-COVID. We therefore did not include this result in the revised

The timing for measuring type I IFN/type I IFN gene signatures would also be difficult on a practical level since patients progress at different speeds and would need to be sampled before reaching their maximal disease severity.

Response: We agree that there could be practical considerations in directly translating our findings to the clinic. In particular, as the reviewer notes, it may not be possible to sample all patients before they

reach peak disease severity. However, in the subset of individuals who can be sampled at the appropriate stage, our prognostic signature could add value. Moreover, our study provides a unique window into the early-stage dynamics of host response to a respiratory virus. As such, we believe this data resource also illuminates the biology of coronavirus infection. In particular, we were able to use this longitudinal dataset to resolve an apparent contradiction between positive and negative associations of type I IFN response with disease severity.

While the overall number of patients in this study is large, the number of mild individuals that progressed to more severe disease is relatively small and the validation cohort only includes two progressors and two non-progressors. Therefore, while the findings from this study are interesting, further validation will be required to confirm that the observed result is generalisable.

Response: We agree that a larger cohort would provide more information. However, we have repeatedly surveyed the literature and not found additional longitudinal studies that fit the requirements for identifying early-stage prognostic signatures in host immune cells. In fact, as far as we know, ours is the only large-scale longitudinal single cell study of COVID-19 that satisfies all the necessary criteria: longitudinal sampling with at least one early-stage sample (Days 1-8), knowledge of disease duration and severity at each time point, age-balanced cohort. Moreover, despite the small size of the validation cohort (as noted, existing studies are not well suited for this analysis) it is notable that our prognostic signature still showed statistically significant predictive power (revised **Figure 3E**, which now includes one additional sample). Lastly, our results are further corroborated by analysis of plasma cytokines and chemokines, which show concordant expression of IFN-alpha (**Figure 3F, G**).

Were all of the patients in this study infected with the same variant of SARS-CoV-2 and is this information available? This is a key factor that could affect the time course of the disease.

Response: A major advantage of our study is that our samples were collected mostly from March to June 2020, before the major diversification of COVID-19 strains. Consequently, the strain diversity of our cohort is likely to be limited. We have indicated in the revised manuscript that the samples were collected early in the pandemic (**Results**, first paragraph).

Is SOCS3 expression enriched overall in T cells and NK cells in severe/critical COVID-19 relative to mild COVID-19? Since DE analysis has been performed in e.g. subpopulations of T cell subsets, it is hard to grasp the overall importance of this finding.

Response: *SOCS3* expression is indeed significantly enriched overall in T and NK cells of severe/critical COVID-19 subjects, and this enrichment peaks after Day 8 (**Supplementary Figure 27A**, see below).

Supplementary Figure 27A: *SOCS3* gene expression (pseudobulk) among all T, NK cells in each sample. Samples are grouped by disease severity and duration. Horizontal dashed line: average *SOCS3* z-score of the 5 asymptomatic (Asy) samples. P-values: Kruskal-Wallis test.

In which cell subsets is *SOCS3* enriched in the scRNA-seq datasets analysed in Figure 7A-D?

Response: **Figure 7A** shows bulk RNA-seq data, and thus cell subsets cannot be identified. We re-analyzed the data in **Figure 7B** (Yoshida et al., scRNA-seq PBMC) one cell type at a time, using cell type labels provided by the authors. The vast majority of individual cell types failed to reach significance in the same ANOVA (FDR Q -value < 0.05, Benjamini-Hochberg correction), and no cell type significantly recapitulated the trend seen in the pseudobulk analysis shown in this figure panel (data not shown). This is perhaps attributable to the fact that the number of samples with active infection containing at least 10 cells of any given cell type was always less than 5 in this dataset, and thus the per-cell-type ANOVA had limited statistical power. For similar reasons, none of the individual cell types gave a significant ANOVA result when we re-analyzed the data in **Figure 7D** (Ziegler et al., scRNA-seq nasal). However, in the re-analysis of data from **Figure 7C** (Yoshida et al., scRNA-seq airway), two cell types did achieve statistical significance: duct and goblet cells (FDR Q -value: 0.037, 0.045) and recapitulate the trend seen in the original airway pseudobulk analysis (**Reviewer Figure 2** see below).

Reviewer Figure 2: Per-cell-type pseudobulk SOCS3 expression in healthy, mild/moderate, severe and convalescent COVID-19 subjects, inferred from upper airway scRNA-seq by averaging across cells within each cell type annotated by (Yoshida et al.). Left: duct cells; Right: goblet cells. Benjamini-Hochberg FDR Q -values are shown in the top left, from Kruskal-Wallis test, one-way ANOVA.

The authors provide no direct evidence that SOCS3 is regulating type I IFN signalling. SOCS3 typically regulates signalling downstream of cytokines using the gp130 receptor, for example IL-6 and OSM. Do the authors have evidence for a non-canonical role of SOCS3 in COVID-19?

Response: As the reviewer accurately noted, our *in vitro* analysis focused on testing the effect of SOCS3 on COVID-19 viral replication, since previous studies have already demonstrated its role in inhibiting type I IFN signalling (Akhtar et al., 2010; Sakai, Takeuchi, Yamauchi, Narumi, & Fujita, 2002). We agree also that SOCS3 is known to negatively regulate IL-6 signalling.

The author's message about the role of SOCS3 is a bit confusing. In Figure 6, SOCS3 is specifically upregulated in subsets T cells and NK cells, while in Figure 7, the authors assess its role in epithelial cells. The mechanism described in epithelial cells may be independent of type I IFNs since they were not measured in this assay. The authors should clarify their line of thinking and validate the inhibition of type I IFN signalling by SOCS3.

Response: Our initial discovery of differential SOCS3 expression was indeed based on single cell analysis of PBMCs. However, in **Figure 7C, D**, we show that SOCS3 is also upregulated in nasal and upper airway epithelial cells of severe/critical COVID-19 subjects. Moreover, type I IFN signalling is active in the HEK-ACE2 epithelial cell line, which has been widely used as a model system for studies of viral infection since it expresses both the receptors required for entry of SARS-CoV-2. For this reason, we used this epithelial cell line for our *in vitro* analysis of the role of SOCS3 in modulating viral replication. We have clarified this point in the revised manuscript.

There are some relatively abundant T cell subsets missing from Figure 1G that are usually easily identifiable by scRNA-seq of PBMCs. These include MAIT cells and V δ 2 $\gamma\delta$ T cells. Are these present within the dataset and if so, where are they located on the UMAP?

Response: MAIT cells are indeed present in our dataset, as are gamma-delta T cells. The dataset also includes other relatively rare cell types, including innate lymphoid cells and double negative T cells. However, as is common in single cell studies of COVID-19 (Bernardes et al., 2020; Kusunadi et al., 2021; Schulte-Schrepping et al., 2020; Su et al., 2020; Szabo et al., 2021), we classified cell types at an

intermediate level of resolution, i.e. we did not attempt a fine-grained classification of all known blood cell types. In this study, our main goal was to identify molecular markers of disease severity. Our major findings in this regard relate to interferon response, which we analysed at single-cell resolution or as single-cell pseudobulk (**Figures 2; 3A-D; 4A-F; 6**), as well as at intermediate and low resolution (**Figure 1D-I; 3B-G; 4G-I; 5; 7**).

Were all T cells included in the clonality analysis performed in Figure 1? Since there are expected differences in clonality between different T cell subsets and the frequency of T cell subsets changes with disease, this analysis should be performed at the level of individual T cell subsets. Alternatively, the authors should show that their analysis is not influenced by the change in proportion of T cell subsets. The same should be done for the B cell analysis.

Response: The clonality analysis in **Figure 1I** (**Figure 1J** in the original manuscript) included all T cells. We note that the expanded T cell clones were primarily contributed by cytotoxic CD8+ T cells (**Supplementary Figure 3C**). We have now repeated this analysis based on cytotoxic CD8+ T cells alone, and reproduced the same result: clonal expansion is still greater in Mild COVID-19 (**Supplementary Figure 3D**). Similarly, we re-analyzed B cell clonality using only memory B cells, and again recapitulated the lack of a clear upward or downward trend with increasing COVID-19 severity (**Supplementary Figure 4C, D**).

Minor comments:

To what stage of disease did the mild patients in the validation cohort progress to? It would be useful to have data from mild patients that developed severe/critical disease.

Response: One mild patient progressed to moderate disease, and one to severe/critical (**Figure 1C**).

Figure 1D is misleading since it suggests that the Mild2 patients all progressed to critical disease, while this is only true for one patient based on Figure 1C.

Response: Figure 1D was intended as a schematic illustration of the concept of a Progressor, rather than an illustration of actual data. However, to avoid confusion on this point, we have now removed Figure 1D.

Figure 1G – have these p-values been corrected for multiple testing? Do any of the other 41 plasma protein markers differ between progressors and non-progressors? i.e. the protein data should be analysed in an unsupervised manner rather than initially selecting proteins that correlate with the expression of the IFN gene signature – this may reveal other useful biomarkers for distinguishing progressors vs. non-progressors.

Response: The above comment on correcting for multiple testing cites **Figure 1G**. However, it probably relates to **Figure 3G**, which shows the analysis of 3 of the 33 plasma protein markers analyzed in **Figure 3F** (in the original manuscript we had indicated that 41 proteins had been tested, but in practice we tested only 33, since 8/41 proteins profiled using the luminex assay failed to pass QC).

As shown in **Figure 3F**, only three protein markers showed high correlation with our mRNA-based 13-gene IFN signature. The other 30 protein markers were not correlated with our prognostic IFN mRNA signature, and we therefore did not use them to differentiate Progressors from Non-Progressors. Due

to the small number of tests (3 tests), we showed the raw p-values for these 3 markers in **Figure 3G**, without correcting for multiple testing. We have now confirmed that, even after correcting for 3 tests (Benjamini-Hochberg), all three proteins are significant at $FDR < 0.05$ (**Reviewer Figure 3**, see below).

To render the analysis in **Figure 3G** independent of that in **Figure 3F**, we have also generated a version of the latter that excludes baseline Progressor samples. The 3 protein markers remain the most highly correlated with our 13-gene IFN signature (**Reviewer Figure 3**, see below).

As suggested by the reviewer, we have now also tested the prognostic power of all 33 protein markers, independently of our RNA-based prognostic signature. The three proteins referred to above (IFN-alpha, MCP-1 and IP-10) were the most significant, and one additional protein, IFN-gamma, had a raw p-value below 0.05. However, after correcting for 33 tests, none of these protein markers showed a significant difference at baseline between Progressors and non-Progressors ($FDR > 0.05$; Benjamini-Hochberg correction). The entire set of 33 raw and corrected p-values is now provided as **Supplementary Table 3**.

Reviewer Figure 3: A. Histogram: expression correlation of 33 plasma proteins with the prognostic 13-gene ISG signature across 73 samples (10 baseline Progressor samples were removed from the set of 83 baseline samples). Red box indicates the three highly correlated proteins: IFN-alpha, MCP-1, IP-10. Scatterplots illustrate the correlation between the three proteins and the prognostic 13-gene

signature across 73 samples. Each dot represents a single sample. Blue lines: linear regression. FDR: Q-values: Benjamini Hochberg correction on linear regression p-values. **B.** Box plots showing the plasma expression scores of the three proteins in 10 Mild Progressors (red) and 37 Mild non-Progressors (blue). FDR q-value: Benjamini-Hochberg correction applied to p-values from Student's t-test (greater).

Figure 4 – do the authors have Luminex data to match Figure 4 i.e. showing the reduction in IFN α over time in progressors and non-progressors? This would further validate the finding.

Response: We thank the reviewer for this suggestion. We have now generated a version of **Figure 4G-I** based on IFN-alpha protein level in plasma (revised manuscript **Supplementary Figure 8**, see below). One of the 11 Mild Progressors is not shown here, due to absence of luminex data. As in **Figure 3G**, this figure shows again that baseline IFN-alpha protein level is high in the plasma of Progressors (filled circles), and that IFN-alpha drops to the background level at the very next sampling (empty circles).

Supplementary Figure 8: Plasma IFN-alpha protein expression z-score of Mild Progressor (circles) and Non-Progressor (box plots) samples, grouped by disease severity and duration. Filled circles: baseline samples of Progressors. Empty circles: second samples of Progressors.

Figure 4G/J/I – how does the metagene analysis performed here differ from the scores in Figure 3B-D? Is it the same analysis, but just showing the results for individual patients and time points? If so, this should be clarified in the text, as it currently implies that a new analysis was performed.

Response: This is indeed the case. **Figure 4G-I** shows the evolution of the 13-gene IFN signature (metagene) over time for all samples, with lines connecting successive data points for Progressors. **Figure 3B-D** shows expression of the same metagene, but only for baseline samples.

Line 292 – the authors should rephrase this sentence since the prognostic gene signature would not always be useful during Days 1-8 since one of the patients progressed during the first four days and another during the 5–8-day period.

Response: We fully agree that this signature does not necessarily persist throughout Days 1-8. We have now revised the sentence as follows:

“Thus, the prognostic type I IFN response signature was transient, and observable only in baseline samples collected during Days 1-8, before the peak of disease severity.”

It would be useful if Supplementary Table 1 could indicate which patients were included in which analyses, since the authors subset the data in different ways throughout the paper. A summary figure displaying this information would also be helpful to the reader.

Response: We appreciate this suggestion and have now updated **Supplementary Table 1** accordingly.

Figure 5 – have the p-values here been corrected for multiple testing?

Response: We have updated the legend to indicate that we performed a non-parametric one-way ANOVA for each cell type (Kruskal-Wallis test), to test the null hypothesis that the means were identical across all disease severity groups. Thus, only one hypothesis test was performed per cell type.

Line 334 – should this be “coloured” rather than “clustered”. Otherwise I am not clear how the authors have clustered the data.

Response: We apologize for the lack of clarity. We have now cited the specific figure panels that show the heatmaps of cells clustered by their fold enrichment trajectories. We have also changed the word “scores” to “trajectories” to indicate the basis for cell clustering.

The top gene is missing from Supplementary Figure 2A.

Response: We thank the reviewer for catching this. We have now indicated the name of the top gene in **Supplementary Figure 2A**.

Reviewer #2 (Remarks to the Author):

General:

In this manuscript, Lin et al describe the dynamic change of the immune cell profiles in response to SARS-CoV-2. The authors conducted a longitudinal analysis, especially focusing on early stages of infection. A total of 286 blood samples was collected from 108 COVID-19 patients and subjected to the single-cell level gene expression analysis. The patterns of TCRs and BCRs were also analyzed. One of the most relevant issues is that the previous studies which has analyzed a large number of longitudinally collected blood samples has been relatively rare so far. In addition to the series of transcriptome analysis, other types of multiple omics analyses were also conducted by microarray-based genotyping analysis, Luminex assays for the level of inflammatory cytokines, bulk RNA-seq and

whole-genome bisulfite sequencing. To validate the deduced hypothesis, the authors also utilized some external cohort datasets. As a result of the integrative analyses of the collected datasets, they found that upregulation of type I IFN-stimulated genes (ISGs) should be the early signature of the subsequent worsening of the patients. There, the ISG expression was dynamic and rapidly receding to the original levels. In contrast, in severe and critical patients, the ISG expressions were deficient and IFN suppressors, such as SOCS3, were upregulated. The in vitro assays also showed that SOCS3 inhibition reduced SARS-CoV-2 replication.

Overall, I fully appreciate that the original blood samples collected for this study should be substantial and valuable. However, since the samples are initially “pooled” and de-multiplexed based on the genomic variations later for the single cell analysis, I have a remaining concern that the produced datasets may have been substantially affected, thus imposing a problem on a precise analysis of the dynamic changes of discrete cell types and the gene expression changes within. Even admitting the analyses may have been conducted in a properly age-/gender-/severity- matched manner, the deduced story about the IFN-SOCS axis remains still controversial.

Response: We are grateful to the reviewer for appreciating the value of this longitudinal data resource. Our study affords a unique longitudinal window into the early stages of the COVID-19 disease trajectory. Now that mild COVID-19 patients no longer present at clinics, it may be challenging for other teams to perform a similar cohort study of the early-stage disease dynamics of COVID-19. Thus, we believe this is a valuable resource for understanding COVID-19 host response and preparing for future pandemics. Given the history of SARS and then COVID-19, it is possible (and perhaps even likely) that a future pandemic could originate from a related pathogen.

Regarding the reliability of the sample pooling strategy, we note that this is in fact a well-established method in the single cell field, adopted by virtually all large single cell consortia worldwide. Sample pooling is used by over a dozen teams within the Human Cell Atlas (for example: CZI-funded Ancestry Networks consortia); the sc-QLgen consortium; the CARDINAL study of >5,000 individuals from UK Biobank; Joseph Powell's OneK1K study and its successor, TenK10K. In addition, a number of published studies are based on this strategy (for example: Perez et al., Science 2022, 376:eabf1970; Yazar et al., Science 2022, 376:eabf3041; Bergamaschi et al., Immunity 2021, 54:1257.e8), and major single cell technology providers have provided their own solutions for sample pooling (for example: 10x Genomics CellPlex kit). Our approach of sample pooling using the genetics-based demuxlet technique is the preferred method in the field when samples are from genetically distinct individuals. This is because genetics-based demultiplexing does not require attachment of sample barcodes to individual cells – the experimental protocol is thus simpler and less error-prone. We lead the Asian Immune Diversity Atlas (AIDA), an 8-nation consortium using single cell omics to profile human immune diversity and its genetic underpinnings. All 8 teams within AIDA use this sample pooling strategy. Far from introducing artifacts, sample pooling is actually the preferred strategy for modern large-scale single cell projects, since it reduces sample-to-sample technical variation by processing multiple samples in a single pool.

Major Comments:

1. To my knowledge, most of the described contents, except for the issues as described above, is confirmatory to or as expected from the previous studies (such as <https://doi.org/10.1016/j.xcrm.2020.100166>).

Response: We thank the reviewer for highlighting the paper by Chevrier et al., which describes a cross-sectional study of COVID-19 based on plasma proteomics and mass cytometry analysis of peripheral immune cells. In contrast, we have longitudinally profiled subjects in our cohort, in most cases with a baseline sample within the first 8 days post symptom onset. Moreover, our analysis is based on single cell RNA-seq, rather than bulk protein profiling. We have now incorporated findings from Chevrier et al. in the Introduction section.

Nevertheless, I consider the presented results should have important implications regarding the IFN-SOCS3 axis. However, I would like to point out that the story is still controversially and should need further extensive analyses before reaching to a particularly conclusion.

Response: We thank the reviewer for highlighting the implications of our study regarding the IFN-SOCS3 axis. As we describe in the Discussion, there have indeed been conflicting reports in the literature on the association between type I IFN and COVID-19 severity. To the best of our knowledge, ours is the first study to resolve the distinctions between early- and mid-stage IFN response, which could contribute to addressing this issue. We have been able to address this question systematically, thanks to our access to a unique longitudinal cohort with early baseline sampling and detailed metadata on disease duration and severity at each time point.

For example;

1-1: How does the longitudinal expression change for the SOCS3 gene or other IFN suppressor genes look like in the same individuals in vivo? Also, the authors should further inspect the behaviors of key genes at the critical time points both at the gene expression levels and at the cytokine levels.

Response: We have now included **Supplementary Figure 8** (see above), which shows that SOCS3 expression in T and NK cells increases (cross-sectionally) with COVID-19 severity at all disease durations, though the increase is most pronounced beyond Day 14.

We also appreciate the importance of cytokine gene expression in this context. As shown in **Figure 2**, all three major PBMC lineages show upregulation of cytokine-mediated signaling genes in Progressors, relative to Non-Progressors. Details of the genes contributing to these cytokine signatures are provided in **Supplementary Table 4**. In **Figure 3F**, we show the correlation of plasma cytokine levels with our 13-gene IFN expression signature in PBMCs, and **Figure 3G** suggests that early-stage protein levels of some key cytokine-related markers (IFN-alpha and the ISGs MCP-1 and IP-10) may be prognostic for COVID-19 severity. In cross-sectional analysis, we show that CD4 Tcm subpopulations expressing *SOCS3* and other negative regulators of IFN response are enriched in severe COVID-19 (**Figure 6B**), while cytotoxic CD8 T cells expressing ISGs are depleted (**Figure 6C**).

1-2: Before conducting such an analysis, I'm concerned that the correlation between the scRNA-seq gene expression levels and their product cytokine levels is generally poor. The authors should further examine possible reasons with this regard.

Response: We would like to clarify that our 13-gene ISG signature did not include any genes with cytokine products (**Figure 2**). While multiple previous studies have investigated circulatory cytokine and chemokine levels in COVID-19, our study was primarily focused on single cell transcriptomic analysis. Nevertheless, as noted above, this 13-gene RNA signature correlates with plasma cytokine levels (**Figure 3F, G**).

1-3: Reasons of different IFN-SOCS3 responses at the early stage are not fully characterized. Were those individuals originally vulnerable for the induction or are there any other reasons? Would not the data or further extensive analysis using the samples collected after the convalescence period, or totally healthy controls, give a clue?

Response: The question of what causes some individuals with mild COVID-19 to have a pronounced type I IFN response early in the disease course is indeed intriguing. One possibility is that the Mild Progressors in our cohort had inherently elevated IFN response. However, this is not likely, since our data indicate that their IFN signature reverts to that of non-Progressors and asymptomatic individuals by Day 14 (**Figure 4G-I**). We agree also that scRNA-seq data from healthy and convalescent samples would provide additional reference points. However, our cohort unfortunately did not include such samples. We obtained access to plasma cytokine and chemokine levels of a small subset of our cohort, but the data were again too limited to observe statistically significant trends (see response to second major comment from Reviewer 1 above).

2. Technically, potential concerns related to the analysis using the “pooled” samples should be fully removed.

Response: As noted above, sample pooling is a well-established method in the single cell field, adopted by virtually all large single cell consortia worldwide. Sample pooling is used by over a dozen teams within the Human Cell Atlas (for example: CZI-funded Ancestry Networks consortia); the sc-QLTgen consortium; the CARDINAL study of >5,000 individuals from UK Biobank; Joseph Powell's OneK1K study and its successor, TenK10K. In addition, a number of published studies are based on this strategy (for example: Perez et al., *Science* 2022, 376:eabf1970; Yazar et al., *Science* 2022, 376:eabf3041; Bergamaschi et al., *Immunity* 2021, 54:1257.e8), and major single cell technology providers have provided their own solutions for sample pooling (for example: 10x Genomics CellPlex kit). Our approach of sample pooling using the genetics-based demuxlet technique is the preferred method in the field when samples are from genetically distinct individuals. This is because genetics-based demultiplexing does not require attachment of sample barcodes to individual cells – the experimental protocol is thus simpler and less error-prone. We lead the Asian Immune Diversity Atlas (AIDA), an 8-nation consortium using single cell omics to profile human immune diversity and its genetic underpinnings. All 8 teams within AIDA use this sample pooling strategy. Far from introducing artifacts, sample pooling is actually the preferred strategy for modern large-scale single cell projects, since it reduces sample-to-sample technical variation by processing multiple samples in a single pool.

2-1: Although I understand there are careful validation analyses in the previous paper (ref 25), further detailed validation should be needed for this particular dataset shown to guarantee the clear demultiplexing of the “pooled” samples based on the genomic polymorphisms. It is desirable that the close relatives or patients of the same ethnic backgrounds are not assigned to the same “pool”, but the genotyping analysis seems to have been done after or independently from the single cell analysis. I also wonder whether the de-multiplexing power should be sufficient for all the cell types, depending on the genomic polymorphisms on the mRNAs expressed on the respective cell types. For example, how many SNPs were utilized for the respective cell types and how discrete each sample was separated?

Response: We performed genetic demultiplexing based on the genotypes of ~387,000 SNPs. We note that Demuxlet is routinely used on pooled samples from donors of the same ethnic background (see

above). In fact, it was originally developed and validated on samples from the same ancestry (Kang et al., Nat Biotechnol 2018). There is no concern regarding use of genetic demultiplexing on PBMC samples – it was in fact first applied to samples of this type. The strategy has subsequently been widely used in studies based on PBMCs. For example: Perez et al., Science 2022, 376:eabf1970; Yazar et al., Science 2022, 376:eabf3041; Bergamaschi et al., Immunity 2021, 54:1257.e8. Note that the latter study is on PBMC samples from COVID-19 patients.

2-2: When the total number of 346,680 cells was divided by a total number of 286 individuals (Line 158), the resulting ~1,200 cell per sample should be too sparse for a current standard single cell RNA seq. This limited number of cells should be particularly problematic when TCR or BCR repertoires are analyzed. For those cases, exact overlap of particular serotypes is not expected between different individuals, thus using as many cells as possible (currently approximately up to 10,000 cells) is desirable for the reasonable analysis. Also, careful evaluation should be needed to confirm the sequencing depth should be sufficient and the representation of the cells from different individuals should be reasonably equal at the sufficient level for this study.

Response: We note that the number of cells per sample (median: 1,105 cells) in our study is well within the norms for large-scale single cell RNA-seq studies of PBMCs. For example: van der Wijst et al., Nat Genet 2018, 50:493 (~550 cells) and Yazar et al., Science 2022, 376:eabf3041 (~1,000 cells). Multiple single cell studies of COVID-19 have also been based on a similar number of cells per sample, including Schulte-Schrepping et al., Cell 2020, 182:1419 (1,350 cells), Zhu et al., Immunity 2020, 53:685 (2,000 cells). We now provide **Supplementary Figure 37** (see below) showing a histogram of the number of cells post-QC from each sample, indicating that the vast majority of samples yielded >500 high-quality cells (see below).

We also note that we have not attempted to analyse overlap of serotypes between different individuals, since this is beyond the scope of our study.

The adequacy of the sequencing depth per cell in our study can be judged by examining the number of detected genes (NODG) per cell. In our case the mode of the NODG in major PBMC cell types ranged from 1,500-1,800 (**Supplementary Figure 30B**) – this is well within the norms for PBMC scRNA-seq data.

Supplementary Figure 37: Histogram of number of high-quality (post-QC) cells per sample.

2-3: I also have a remaining concern to what extent the batch effects may have been compromised by the “pooling” approach.

Response: As alluded to above, a major advantage of the pooling strategy is that it reduces batch effects in cohort studies, since multiple samples are processed simultaneously in a single batch.

Because of these reasons, at least, for some representative samples, a careful validation analysis should be conducted using individually prepared libraries.

Response: We respectfully disagree that the sample pooling strategy requires validation. As noted above, it is a well-established method in the single cell field, and in fact considered superior to processing each sample individually.

3. It seems to me that the produced genotype and methylation type datasets, which could deepen the current study, are not fully utilized. In addition, it is a pity that methylation analysis is for a different group of patients. In fact, much more information could be extracted if the methylation analyses should be directly conducted for the same individuals used for the single cell analysis.

Response: The data on SOCS3 methylation were kindly provided by the authors of a different study - they will report a more detailed analysis of whole-genome methylation in their own paper (Kumar et al., *submitted*).

Minor Comments:

4. Figure 1: Although the authors annotated T cells and NK cells into detailed cell types in Figure 1, hereafter, they compared gene expression by four major cell types only. The authors should comment on whether the profiles are not different between those finer sub-classes.

Response: We apologize for the lack of clarity on this point. Our differential expression analysis of Progressors vs. non-Progressors, which was used to define the 13-gene type I IFN signature, was not performed on the four major cell types. Rather, as described in Lines 190-191 (original manuscript), we compared gene expression of finer sub-classes, and then subsequently merged the sub-class-specific DEG sets by major cell type, since the prognostic signature was highly consistent across sub-classes.

In this study, we have classified cells at multiple levels of resolution, as is common in scRNA-seq studies. While some analyses benefit from the greater statistical power of grouping cells into major lineages (T and NK, B, myeloid), we have also analysed the data at the highest possible level of resolution: individual cells. For example, in **Figure 2A-C**, we show cellular-resolution analysis of cell states enriched in Progressors and interpret these based on their membership in finer sub-classes such as CD16bright NK, Cytotoxic CD8 T, CD4 Tem, Naïve CD8 T, Memory B and CD16 Mono. **Figures 3 and 4** again include cellular-resolution views of the IFN signature score and temporal dynamics, respectively. **Figure 5** is entirely based on finer sub-classes, and **Figure 6** includes all levels of resolution from major lineage to individual cell. Overall in the manuscript text, we have drawn attention to finer sub-classes wherever the results indicated a distinct pattern.

5. Figure 1B: It seems that the samples are biased to symptomatic patients (only five individuals are

recruited for “asymptomatic” cases). At least recently, is not it so difficult to recruit asymptomatic patients at the asymptomatic stage?

Response: We agree with the reviewer that a larger cohort could potentially provide more information. In this case, since our cohort was recruited in March-June 2020, before mass testing had been established in Singapore, the number of asymptomatic subjects was limited.

6. Figure 1C: In fact, the patients recruited for the current study showed various pattern of symptom patterns. As indicated by the authors, the variable host immune response to SARS-CoV-2 should be the key. Some of those response may accompany global changes on cellular populations. I wonder whether those factors can be analyzed from “pooled” samples as well. Even though the single cell analysis could not be conducted for this large number of samples, there should be some surrogate way, such as FACS or CyTOF, to confirm the global populations.

Response: In **Figure 5**, we do indeed show abundance shifts in specific sub-classes of T and NK cells, B cells and myeloid cells (see also **Supplementary Figures 21** and **22**). We note that sample pooling does not impair our ability to analyse changes in cell population abundance – once the cells are demultiplexed, the data can be analysed exactly as one would analyse conventional scRNA-seq data (cells are bioinformatically assigned to their respective samples, as described in Kang et al., Nat Biotechnol 2018).

We thank the reviewer for raising the question of other cellular-resolution assays such as FACS and CyTOF. In the manuscript text, we have compared our results to previous FACS- and CyTOF-based studies of COVID-19, for example: Ren et al., Cell 2021; Lee et al., Science Immunology 2020; Schulte-Schrepping et al., Cell 2020; Matthew et al., Science 2020 and Kuri-Cervantes, Science Immunology 2020.

7. To elucidate the T cell response, the authors mainly analyzed the top ten clonotypes. It is unlikely that the key T cell population should consist such a large population, particularly at the early response, the analytical strategy should be re-considered to separate TCR patterns related to severity and/or progression. It is desirable to also conduct the repertoire sequencing in bulk and compare the results with each other.

Response: To clarify, **Figure 11** does not show the early-stage TCR response to COVID-19. Rather, it shows the response from Day 9 onwards. To address this point regarding the number of relevant TCR clonotypes, we have now re-analyzed the data based on the top 5 TCR clones, rather than the top 10, and updated **Figure 11** and **Supplementary Figure 3B** accordingly (see revised figure below). Our conclusion remains the same.

We agree with the reviewer that it would be interesting to examine the specific TCR clones that relate to COVID-19 severity and/or progression, and attempt to infer the epitopes they target. However, this would require a separate assay that focused on TCR repertoire profiling of 10,000-100,000 T cells, which is beyond the scope of this study focused on single cell transcriptomics.

Revised Figure 11: T cell clonality index (fraction of T cells derived from the 5 most abundant TCR clones; Day 9 onwards), estimated using the single cell immune profiling assay. p-value: Kruskal–Wallis test.

Revised Supplementary Figure 3B: T cell clonality index (fraction of T cells derived from the 5 most abundant TCR clones; Days 1-8), estimated using the single cell immune profiling assay. p-value: Kruskal–Wallis test.

8. Figures 2 and 6 are not always informative, thus, are not necessary. At least, they could be presented in a far more concise manner.

Response: **Figure 2** shows the one of the central results of this study, namely how the prognostic early-stage type I IFN signature was identified. It contains three components: 1) the cellular-resolution map of cell states enriched in progressors relative to non-progressors, which we subsequently compare to the type I IFN-expressing cell states in **Figure 3B-D**; 2) the GO-term analysis that initially established type I IFN response as the strongest prognostic pathway and 3) the expression level of each prognostic

type I IFN gene in each Progressor or non-Progressor sample. We agree with the reviewer that these three components are repeated a number of times in the figure, to cover two temporal ranges (Days 1-4 and Days 5-8), Mild1 as well as Mild2, and the three major cell lineages. Strikingly, in every one of these analyses, type I IFN appears as the top GO term among prognostic markers, which underscores the centrality of this pathway. Nevertheless, we take the reviewer's point that it may seem repetitive to display every one of these results as a main figure. We have now simplified **Figure 2** by moving the Mild2 results (**Figure 2D-F**) to **Supplementary Figure 6**.

Figure 6 is less repetitive, since it regresses out disease duration and focuses on a single disease severity class. Cell lineage (T and NK, B, myeloid) is the only axis of repetition in this figure. We could potentially move one of the three cell lineages to Supplementary Material. However, since each lineage has its own paragraph in main text (and sometimes two paragraphs), we hope it would be acceptable to retain all three lineages in **Figure 6**.

9. The term used in the manuscript should be unified; for example, SARS-CoV-2 (line71) and SARS-CoV2 (line 475 and line 477).

Response: We thank the reviewer to detecting this inconsistency. We have amended the text to use consistent terminology (SARS-CoV-2).

10. Fig 1C: In the main text, the authors describe the number of mild1and mild2 progressors only. Judging from the Figure, there seems also progressed patients in moderate and severe groups. Are there any particular difference between them?.

Response: As described in response to a related question from Reviewer 1, we have now examined the baseline 13-gene IFN signature during Days 1-8 of the 2 Moderate Progressors and 4 Moderate Non-Progressors in our cohort. The IFN signature is indeed highest in one of the two Moderate Progressors – this individual progressed to Severe disease subsequently. However, due to variability across patients and the small number of individuals (6), the difference between Moderate Progressors and Non-Progressors was not statistically significant (p-value=0.255; **Reviewer Figure 1**, see response to Reviewer 1 above).

We also examined the 13-gene IFN signature in the 3 Severe Progressors vs. 2 Severe Non-Progressors in our cohort (Days 1-8). Again, the number of individuals in this category was too small to detect a signal (p-value>0.05, data not shown).

11. Fig 1E: The authors grouped PBMCs into seven major groups. For dendric cell, they only show cDC. However, previous scRNA-seq studies, using PBMC for COVID-19 analysis, generally separate pDC and cDC or collectively annotated them as DC. Is there any specific reason to depict only cDC for the annotation?

Response: In this study, given the cohort size (286) and the number of post-QC cells per sample (median: 1,105), we did not expect to be able to draw conclusions regarding prognostic or cross-sectional expression signatures of rare cell types such as pDCs. We therefore clustered cells based on genes expressed in at least 1% of cells. This gene filter reduces noise from lowly expressed genes, but precludes detection of rare-cell clusters such as pDCs. As noted above, previous single cell studies of COVID-19 have similarly focused on more common cell types, for the same reason.

12. Fig 2: Although authors annotated cells into subtypes in Fig 1G- 1I, they identified DEGs based on major three cell types, T/NK, B and Myeloid cells. Even though the result demonstrated that the type I interferon signaling pathway is the common feature in all cell types, this method doesn't seem to take advantage of scRNA-seq. On the other hand, in Fig 6, they identified severity associated gene ontologies based on the detailed cell type. I wonder if there are any reasons for changing the analytical axes for the respective analyses?

Response: As noted in our response to Minor Comment 4 from the same reviewer, we did not identify DEGs based on the three major cell types. Rather, as described in Lines 192-193, we compared gene expression of finer sub-classes, and then subsequently merged the sub-class-specific DEG sets by major cell type (**Figure 2**), since the prognostic signature was highly consistent across sub-classes.

In this study, we have used multiple levels of resolution in grouping cells together. While some analyses benefit from the greater statistical power of grouping cells into major lineages (T and NK, B, myeloid), we have also analysed the data at the highest possible level of resolution: individual cells. For example, in **Figure 2 A-C**, we show cellular-resolution analysis of cell states enriched in Progressors and interpret these based on their membership in finer sub-classes such as CD16bright NK, Cytotoxic CD8 T, CD4 Tem, Naïve CD8 T, Memory B and CD16 Mono. As noted above, we also perform DEG analysis at the level of cell sub-classes. **Figures 3** and **4** again include cellular-resolution views of the IFN signature score and temporal dynamics, respectively. **Figure 5** is entirely based on finer sub-classes, and **Figure 6** includes all levels of resolution from major lineage to individual cell. Overall in the manuscript text, we have attempted to draw attention to finer sub-classes wherever the results indicated a distinct pattern.

Reviewer #3 (Remarks to the Author):

The authors performed a longitudinal single cell RNA-seq on 286 blood samples from age- and gender-matched COVID-19 patients, including 73 with early samples. The data presented in this work are extremely informative as the longitudinal design allows evaluating the response profile in patients by dividing them on the basis of the different course of the disease. The main result of the work is the description of the trend of the type1-IFN response in relation to the disease worsening. Differently from the results published in several papers (cited by the authors in discussion), the data presented here demonstrate a high interferon response in patients who subsequently worsen and show as this response is associated with the concentration of two plasmatic mediators associated with severe disease (IP-1 and MCP-1). There are some points that need clarification.

Response: We are grateful to the reviewer for highlighting the novelty of this longitudinal single cell data resource for COVID-19 host response dynamics, which has allowed us to resolve complex temporal associations of type I IFN response with disease severity.

1) As well described in the discussion, a large literature reports association between a strong Type-I IFNs response and a mild COVID-19 disease. The data presented here support the opposite view, at least in the early stages of the disease. Can the timing 1-8 days from the onset of symptoms represent a good definition of “early response”? This is a long time frame including innate and adaptive immune induction.

Response: We apologize for the lack of clarity on this point. As we describe in the revised manuscript, we defined early response as the period preceding clinical deterioration. In 1,168 COVID-19 patients with moderate disease, the median time to deterioration was 11 days after symptom onset, with an interquartile range of 9-14 days (Chen et al., 2020). Consistently, another study found that chest x-ray anomalies peaked at Days 10-12 post symptom onset (Wong et al., 2020). We therefore defined 1-8 days as the window of early response. Our scRNA-seq data indicate that type I IFN response recedes to the level of asymptomatic subjects by Days 9-14 (**Figure 4G-I**), further supporting the relevance of this definition. The two citations above have now been added to the manuscript.

2) Figure 2: The authors showed an increased expression of ISG in progressor Mild 1 patients both at days 1-4 and at days 4-8. In contrast, this association in Mild 2 patients was only true at days 4-8. How do they explain this difference?

Response: In Figure 2, we did not show cell state enrichment results for Mild2 patients during Days 1-4, since our cohort included only one Mild2 Progressor with a baseline sample in this time window. In other words, the absence of figure panels for Mild2 patients during Days 1-4 in Figure 2 does not imply that ISG expression was not prognostic in this patient group. Rather, it merely indicates that the test was not performed (due to lack of a sufficient number of samples).

3) The authors identify 13 genes associated with the IFN response that are higher in patients who will get worse than in those who will remain stable. The data is certainly interesting but should be validated on a cohort wider than that shown here (4 total patients, 2 progressor and 2 non progressor). Please, add data about a larger validating cohort.

We greatly appreciate the importance of validating our results in an independent cohort. To this end, we performed a systematic survey of 25 existing single-cell RNA-seq studies of COVID-19 and examined their suitability for identifying early-stage prognostic signatures of disease progression. The key requirements in this context are: 1) longitudinal scRNA-seq data, including 2) a baseline sample within Days 1-8, and knowledge of 3) disease duration and 4) disease severity at each time point. Only 3/25 studies satisfied all four requirements. One of these three studies did not include Mild patients (Krämer et al., 2021), and another included only 4 patients who met all the criteria (Liu et al., 2021). The third study is the one we used to validate our ISG signature (Schulte-Schrepping et al., 2020). The data from this study included 2 Mild Progressors and 3 Mild Non-Progressors with baseline samples during Days 1-8. In **Figure 3E**, we showed the expression of the 13 ISGs in 4 of these 5 samples – the fifth was accidentally left out. We have now **revised Figure 3E** to include the fifth sample (see below). Inclusion of this additional sample further strengthens the validation of our 13-gene ISG signature (p -value=0.012 in revised manuscript vs p -value=0.037 in the original version).

Revised Figure 3E: Heatmap shows the scaled pseudobulk expression levels (expression z-score) of the 13 prognostic ISGs in baseline samples from 2 Mild Progressors and 3 Mild non-Progressors in the German cohort. *: p -value ≤ 0.05 , one-sided Student's t-test for upregulation in Progressors relative to non-Progressors. p -value above heatmap: difference between Progressors and non-Progressors in trimmed mean of the 13 genes (Student's t-test, one-sided).

4) They defined the average normalized expression of these 13 genes at baseline as a prognostic score; please, specify baseline, is it day 1-8?

Response: Again, we apologize for the lack of clarity in the text. We have now specified that baseline samples were collected at presentation within Days 1-8.

5) In figure 4G the authors showed a comparison of the expression of the 13 genes between different clinical severities. Levels of these genes in "moderate and mild 2" patients were higher in early times (0-4 and 4-8 days) than in "severe" and "mild 1" patients. Indeed, the figure shows that severe / critical patients showed levels of those 13 genes similar to those observed in mild patients, making difficult the hypotheses of an association between the severity and expression of these genes at T 5-8. To address this problem, the authors suggest that this is the main limitation of cross-sectional studies, while a longitudinal approach overtime allows identifying a higher ISG expression with a high risk of worsening. This aspect represents the main limitation in the use of the set of 13 genes proposed as an early marker of worse prognosis in the clinical management of COVID-19 patients. How could this system be exported to clinical practice? How to define in a single patient whether the early ISG value is decreasing or is stable? The wide timing defined "early" (days 1-8) makes it even more complicated. Please, add a comment on this point.

Response: We agree with the reviewer that the dynamics of IFN response in host immune cells of COVID-19 patients is temporally dynamic, and also correlates with disease severity, both current and future. Indeed, the complexity and variable nature of type I IFN response is one of our major findings. Our longitudinal single cell study of 286 scRNA-seq datasets from 108 COVID-19 patients is, to the best of our knowledge, the only current resource that can separate the effects of disease duration and severity, particularly in the early stages of infection. As noted above, we defined the early stage of infection as the period before clinical deterioration, which is typically observed from Days 9-14. Thus, we defined the early stage as Days 1-8 post symptom onset. This definition is consistent with our molecular findings: ISG expression in our cohort dropped to almost to asymptomatic levels by Days 9-14, and subsided completely beyond Day 14 (**Figure 4G-I**). Our prognostic signature is applicable to the baseline samples (Days 1-8) of Mild1 and Mild2 patients.

Regarding clinical applications, we would like to clarify that, for prognosis, we do not need to know whether the IFN signature is decreasing or stable. We only need to know the IFN signature at a single time point, namely at baseline. All of our prognostic analyses were performed on the set of baseline samples. Our results suggest that patients who present with mild symptoms during Days 1-8 and subsequently worsen have a higher IFN signature at baseline than those whose symptoms remain stable or decline. Thus, the ISG signature at baseline could be used as a prognostic indicator of future disease severity.

As the reviewer accurately notes, we have largely binned Days 1-8 (or Days 1-4 and Days 5-8) together. We fully agree that even greater temporal resolution would be beneficial. However, at this cohort size (108 individuals), it was challenging to obtain insights at higher temporal resolution. To the best of our knowledge, this is the largest longitudinal single cell transcriptomic resource on early-stage COVID-19.

6) In lines 205-207, the authors summarize that the 13 genes discriminate well the mild patients that will get worse from those that will not get worse. Why the authors did not show the analysis of these 13 genes even in moderates who become severe or critical. From figure 1C it is clear that these type of patients were also included in the study? The question is: is the early high production of type-1 IFNs able to predict a worsening only in patients starting with an initial mild disease?

Response: This point is well taken. In the original manuscript, we focused on progression from Mild1 and Mild2, since the cohort included a substantial number of Mild Progressors (11 individuals) and Mild Non-Progressors (52) with baseline samples within Days 1-8. In contrast, we had limited statistical power to differentiate Moderate Progressors (2 individuals within Days 1-8) from Moderate Non-Progressors (4 individuals). We have now examined the 13-gene IFN signature for these 6 individuals with moderate disease at presentation. The ISG signature is indeed highest in one of the two Moderate Progressors – this individual progressed to Severe disease subsequently. However, due to variability across patients and the small number of individuals (6), the difference between Moderate Progressors and Non-Progressors was not significant (p -value=0.255; **Reviewer Figure 1**, see response to Reviewer 1 above).

7) The data about SOCS expression are very interesting and can explain the reduced IFN- α expression and function in severe clinical presentation. At what time point the expression of SOCS was increased? Is this timing compatible with the ISG reduction described in the longitudinal analysis?

Response: To address this point, as well as a similar question from Reviewer 2, we have now included **Supplementary Figure 27A** (see below), which shows that *SOCS3* expression in T and NK cells increases (cross-sectionally) with COVID-19 severity at all disease durations. Elevated *SOCS3* expression beyond Day 14 may not relate to ISG expression levels, since ISGs drop fully to asymptomatic levels by this time (**Figure 4G-I**).

Supplementary Figure 27A: *SOCS3* gene expression (pseudobulk) among all T, NK cells in each sample. Samples are grouped by disease severity and duration. Horizontal dashed line: average *SOCS3* z-score of the 5 asymptomatic (Asy) samples. *P*-values: Kruskal-Wallis test.

Why the author did not include *SOCS* expression in the longitudinal analysis? Can the author verify the possible contribution of *SOCS* expression in defining progressor vs non progressor patients? Is the *SOCS* expression associated with a reduction of plasmatic IFN- α ?

Response: We have now examined *SOCS3* expression at baseline in the 11 Progressors vs. 52 Non-Progressors. We did not observe a significant difference between these two patient groups (**Reviewer Figure 4**, see below). Thus, *SOCS3* appears to be more relevant in cross-sectional, than longitudinal analysis. This result again highlights the value of our dataset in disentangling temporal and cross-sectional dimensions of SARS-CoV-2 host response.

Reviewer Figure 4: Left panel: boxplot of the z-score of pseudobulk (averaged across T, NK cells in one sample) *SOCS3* expression in Progressors (n=11) vs Non-Progressors (n=52). *P*-value: Student's *t*-test (two-sided). Middle, right panels: similar boxplots for myeloid cells and B cells, respectively.

As shown in **Reviewer Figure 5** below, we do not observe a correlation between *SOCS3* pseudobulk expression and plasma IFN-alpha protein levels.

Reviewer Figure 5: Scatterplot of plasma IFN-alpha protein level vs. pseudobulk *SOCS3* expression in PBMCs across 83 paired samples.

8) Linea 99-102. Several cell populations are described in severe COVID-19 and some of them can have a prognostic value in the further worsening of the pathology. Among these, the authors forgot to include the myeloid derived suppressor cells (MDSC) that expand early after infection and can predict a fatal outcome (doi: 10.1038/s41419-020-03125-1).

Response: We fully agree with the reviewer that MDSCs represent an interesting population in this context. As we now note in the revised Introduction, it has been shown that expansion of polymorphonuclear (granulocytic) MDSCs is associated with adverse outcome in COVID-19 (Sacchi et al., 2020). However, this population cannot easily be detected in our study, since PBMC isolation excludes the vast majority of granulocytes, and freeze-thaw tends to damage those that remain.

References

- Akhtar, L. N., Qin, H., Muldowney, M. T., Yanagisawa, L. L., Kutsch, O., Clements, J. E., & Benveniste, E. N. (2010). Suppressor of cytokine signaling 3 inhibits antiviral IFN-beta signaling to enhance HIV-1 replication in macrophages. *J Immunol*, *185*(4), 2393-2404. doi:10.4049/jimmunol.0903563
- Bernardes, J. P., Mishra, N., Tran, F., Bahmer, T., Best, L., Blase, J. I., . . . Rosenstiel, P. (2020). Longitudinal Multi-omics Analyses Identify Responses of Megakaryocytes, Erythroid Cells, and Plasmablasts as Hallmarks of Severe COVID-19. *Immunity*, *53*(6), 1296-1314.e1299. doi:10.1016/j.immuni.2020.11.017
- Chen, S. L., Feng, H. Y., Xu, H., Huang, S. S., Sun, J. F., Zhou, L., . . . Fang, M. (2020). Patterns of Deterioration in Moderate Patients With COVID-19 From Jan 2020 to Mar 2020: A Multi-Center, Retrospective Cohort Study in China. *Front Med (Lausanne)*, *7*, 567296. doi:10.3389/fmed.2020.567296
- Krämer, B., Knoll, R., Bonaguro, L., ToVinh, M., Raabe, J., Astaburuaga-García, R., . . . Nattermann, J. (2021). Early IFN- α signatures and persistent dysfunction are distinguishing features of NK cells in severe COVID-19. *Immunity*, *54*(11), 2650-2669.e2614. doi:10.1016/j.immuni.2021.09.002
- Kusnadi, A., Ramírez-Suástegui, C., Fajardo, V., Chee, S. J., Meckiff, B. J., Simon, H., . . . Ottensmeier, C. H. (2021). Severely ill COVID-19 patients display impaired exhaustion features in SARS-CoV-2-reactive CD8(+) T cells. *Sci Immunol*, *6*(55). doi:10.1126/sciimmunol.abe4782
- Liu, C., Martins, A. J., Lau, W. W., Rachmaninoff, N., Chen, J., Imberti, L., . . . Tsang, J. S. (2021). Time-resolved systems immunology reveals a late juncture linked to fatal COVID-19. *Cell*, *184*(7), 1836-1857.e1822. doi:10.1016/j.cell.2021.02.018
- Patel, M. A., Knauer, M. J., Nicholson, M., Daley, M., Van Nynatten, L. R., Martin, C., . . . Fraser, D. D. (2022). Elevated vascular transformation blood biomarkers in Long-COVID indicate angiogenesis as a key pathophysiological mechanism. *Mol Med*, *28*(1), 122. doi:10.1186/s10020-022-00548-8
- Sacchi, A., Grassi, G., Bordoni, V., Lorenzini, P., Cimini, E., Casetti, R., . . . Agrati, C. (2020). Early expansion of myeloid-derived suppressor cells inhibits SARS-CoV-2 specific T-cell response and may predict fatal COVID-19 outcome. *Cell Death Dis*, *11*(10), 921. doi:10.1038/s41419-020-03125-1
- Sakai, I., Takeuchi, K., Yamauchi, H., Narumi, H., & Fujita, S. (2002). Constitutive expression of SOCS3 confers resistance to IFN-alpha in chronic myelogenous leukemia cells. *Blood*, *100*(8), 2926-2931. doi:10.1182/blood-2002-01-0073
- Schulte-Schrepping, J., Reusch, N., Paclik, D., Baßler, K., Schlickeiser, S., Zhang, B., . . . Sander, L. E. (2020). Severe COVID-19 Is Marked by a Dysregulated Myeloid Cell Compartment. *Cell*, *182*(6), 1419-1440.e1423. doi:10.1016/j.cell.2020.08.001
- Su, Y., Chen, D., Yuan, D., Lausted, C., Choi, J., Dai, C. L., . . . Heath, J. R. (2020). Multi-Omics Resolves a Sharp Disease-State Shift between Mild and Moderate COVID-19. *Cell*, *183*(6), 1479-1495.e1420. doi:10.1016/j.cell.2020.10.037
- Szabo, P. A., Dogra, P., Gray, J. I., Wells, S. B., Connors, T. J., Weisberg, S. P., . . . Farber, D. L. (2021). Longitudinal profiling of respiratory and systemic immune responses reveals myeloid cell-driven lung inflammation in severe COVID-19. *Immunity*, *54*(4), 797-814.e796. doi:10.1016/j.immuni.2021.03.005
- Wong, H. Y. F., Lam, H. Y. S., Fong, A. H., Leung, S. T., Chin, T. W., Lo, C. S. Y., . . . Ng, M. Y. (2020). Frequency and Distribution of Chest Radiographic Findings in Patients Positive for COVID-19. *Radiology*, *296*(2), E72-e78. doi:10.1148/radiol.2020201160

REVIEWER COMMENTS

Reviewer #1 (Remarks to the Author):

Major points:

Overall, while the data is intriguing, I remain unconvinced of the usefulness or reliability of increased type I IFN gene expression as a biomarker of individuals that will progress to more severe disease, particularly in the context of progression to clinically relevant disease e.g. severe disease. For example, in Reviewer Figure 1, only one of the two moderate individuals that progressed has high ISG expression, while the other is comparable to non-progressors. An association between early ISG expression and disease progression will need to be validated in multiple larger cohorts.

Reviewer Figure 4 and 5 should be included and discussed in the main manuscript. It is an important point that SOCS3 expression did not differ between progressors and non-progressors and is also not correlated with IFN α expression. The authors argue that SOCS3 might be an important suppressor of IFN signalling enriched in severe COVID-19. However, differences in ISG expression between progressors and non-progressors are not associated with differences in SOCS3 expression.

Z-scores can exaggerate differences between individuals. It would be helpful if the authors could include plots showing the absolute expression of each gene within the 13-gene signature in each individual, so that the reader can get a better idea of variability in expression of these genes across progressors and non-progressors.

Highly clonal T cell populations (e.g. MAIT cells) should be excluded from the clonality analysis. These could significantly impact the results, since e.g. MAIT cell frequency is reduced in severe relative to mild COVID-19 – loss of these cells in more severe disease would therefore lead to reduced overall T cell clonality.

Minor points:

The fold change of SOCS3 expression in each cell subset for each donor (scRNA-seq data) should be shown e.g. in a box plot.

Line 233 – if 8/13 ISGs were individually enriched in progressors vs. non-progressors, does the predictive score of the ISG signature improve if only these 8 genes are included? Is this only true for the validation cohort?

The paragraph starting line 288 does not make it clear that this analysis is using the same metagene score as earlier in the manuscript, but this time tracking expression by each individual over the disease course.

While I understand that it can be useful to classify cells at an intermediate resolution for scRNA-seq, I believe that it is important to segregate out certain cell subsets that exhibit distinct dynamics to conventional T cells in viral infections, including COVID-19 e.g. MAIT cells.

What is the prognostic score for IFN α protein relative to the 13 gene ISG signature (since you show a correlation in Fig. 3F)? IFN α would be easier to measure clinically than a gene signature.

Rather than testing the 33 Luminex proteins individually for differential expression at baseline between progressors and non-progressors, have the authors tested whether combinations of these proteins can predict disease progression, and how the accuracy/sensitivity of these protein signatures compares with the ISG metagene?

Limitations regarding clinical use of the ISG signature as a biomarker of disease progression should be discussed.

Reviewer #2 (Remarks to the Author):

First of all, I appreciate the substantial additional work which the authors have made for the revision. Thanks to the extensive analyses and the related discussion, I consider the

manuscript has been very much improved. Especially, I appreciate the kind explanation on making use of the “pooled” samples. Thanks to their detailed explanation, I understand its reliable application. I also appreciate the extensive analyses on SOCS3. Even though some issues still remain to be further scrutinized, I understand that part should take time and would delay the publication of this manuscript in a timely manner, which I think is more important. Even now, there are a large number of patients who are newly infected or suffering from long-term symptoms. New viral strains are still emerging. I sincerely hope further in-depth analysis of the collected rich data should lead to better understandings of the mechanism of the viral infection

Reviewer #4 (Remarks to the Author):

The revised version of the manuscript by Lin et al. is very interesting. The study is well-performed and the cohort of patients is quite wide collecting samples from early in disease. This work is supported by a large number of patients monitored throughout the disease course. The longitudinal study allows to determine possible biomarkers for prognostic purpose.

The responses provided to the Reviewer’s comments are satisfactory and have addressed the raised concerns; however, with regard to Response 2, I would suggest to add in the text that the cell state enrichment results for Mild 2 patients were not shown in Figure 2 because of the limited number of patients available to study.

REVIEWER COMMENTS

Reviewer #1 (Remarks to the Author):

Major points:

Overall, while the data is intriguing, I remain unconvinced of the usefulness or reliability of increased type I IFN gene expression as a biomarker of individuals that will progress to more severe disease, particularly in the context of progression to clinically relevant disease e.g. severe disease. For example, in Reviewer Figure 1, only one of the two moderate individuals that progressed has high ISG expression, while the other is comparable to non-progressors. An association between early ISG expression and disease progression will need to be validated in multiple larger cohorts.

We appreciate the point made by the reviewer that our prognostic 13-gene ISG signature is applicable only to patients who present as Mild1 or Mild2 at baseline. We did not have sufficient statistical power to evaluate the prognostic value of this signature in patients who classified as Moderate at baseline, since our cohort included only two Moderate Progressors. Nevertheless, we note that the Mild2 Progressors in our cohort progressed to a condition requiring supplementary oxygen (defined as Moderate in our classification).

We have now updated the Results text to clarify that we did not have sufficient statistical power to test for prognostic signatures in patients who presented as Moderate (i.e. who already required supplementary oxygen).

As we describe in the manuscript, our 13-gene prognostic signature was validated in an independent German cohort. We also corroborated this signature through identification of IFN-alpha and two IFN-related proteins (MCP-1, IP-10) as prognostic markers in plasma. In the revised Discussion section, we now indicate that the prognostic value of our signature should be further validated in larger cohorts, as suggested by the reviewer:

“Further studies in larger cohorts are needed to examine the clinical implications of this signature, particularly for predicting progression to severe and critical COVID-19.”

Reviewer Figure 4 and 5 should be included and discussed in the main manuscript. It is an important point that SOCS3 expression did not differ between progressors and non-progressors and is also not correlated with IFN α expression. The authors argue that SOCS3 might be an important suppressor of IFN signalling enriched in severe COVID-19. However, differences in ISG expression between progressors and non-progressors are not associated with differences in SOCS3 expression.

As suggested, we now include **Reviewer Figure 4** as **Supplementary Figure 27B** in the latest version of the manuscript. We have shown 13 ISGs show higher expression at baseline in Progressors relative to non-Progressors signature (**Figure 3B-D**), potentially driven by higher plasma IFN α levels in Progressors (**Figure 3G**). In this scenario, there is no expectation that SOCS3 will necessarily differ between Progressors and non-Progressors. Thus, **Reviewer Figure 4** does not in any way contradict our key conclusion regarding the prognostic value of the 13-gene ISG signature.

Correlation with IFN α protein expression: SOCS3 is a known modulator of response to IFN α (Sakai et al., Blood 2002, 100: 2926; Carow and Rottenberg, Front Immunol 2014, 5:58). Consistently, our cross-sectional analysis shows that naïve CD4 and CD8 T cells, as well as CD4 Tcm and Tem cells show reduced ISG expression with increasing severity, and simultaneously show increased SOCS3 expression (**Figure 6A, Supplementary Figure 24**). Based on this result, we hypothesized that, in

COVID-19, *SOCS3* upregulation contributed to suppression of ISG expression in severe COVID-19. We note that this mechanism does not require any change in extracellular IFN α expression, since *SOCS3* acts intracellularly to suppress response to IFN α . Thus, **Reviewer Figure 5** does not contradict our results or conclusions in this regard.

Z-scores can exaggerate differences between individuals. It would be helpful if the authors could include plots showing the absolute expression of each gene within the 13-gene signature in each individual, so that the reader can get a better idea of variability in expression of these genes across progressors and non-progressors.

We now provide the expression levels of the 13 genes in each individual in **Supplementary Table 7**.

Highly clonal T cell populations (e.g. MAIT cells) should be excluded from the clonality analysis. These could significantly impact the results, since e.g. MAIT cell frequency is reduced in severe relative to mild COVID-19 – loss of these cells in more severe disease would therefore lead to reduced overall T cell clonality.

To examine the potential contribution of MAIT cells to the reduced clonality of circulating T cells in severe and critical COVID-19, we discarded all cells expressing the MAIT cell marker *TRAV1-2*, and repeated the analysis in Figure 1I on the remaining (non-MAIT) T cells. Since MAIT cells are rare, we obtained a very similar result, thus confirming that the reduced clonality of T cells in severe and critical COVID-19 cannot be attributed to a reduction in MAIT cell abundance (**Reviewer Figure Z1**, see below).

Reviewer Figure Z1. T cell clonality index (fraction of T cells derived from the 10 most abundant TCR clones; beyond Day 8) after removing MAIT cells from all subjects/samples, estimated using the single cell immune profiling assay. p-value: Kruskal–Wallis test. Repeat of **Figure 1I**, with MAIT cells removed.

Minor points:

The fold change of *SOCS3* expression in each cell subset for each donor (scRNA-seq data) should be shown e.g. in a box plot.

We have now repeated Supplementary Figure 27A with expression fold-change, rather than expression z-score, on the Y-axis. Fold change is calculated relative to the mean of *SOCS3* expression across asymptomatic patients (**Reviewer Figure Z2**, see below). We restricted this analysis to T and

NK cells, since we did not observe evidence of increased *SOCS3* expression in B cells, or in Myeloid cells (Figure 6D-I).

Reviewer Figure Z2. *SOCS3* expression fold-change (pseudobulk) among all T, NK cells in each sample, relative to the average of the 5 asymptomatic (asym) samples (horizontal dashed line). Samples are grouped by disease severity and duration. *P*-values: Kruskal-Wallis test.

Line 233 – if 8/13 ISGs were individually enriched in progressors vs. non-progressors, does the predictive score of the ISG signature improve if only these 8 genes are included? Is this only true for the validation cohort?

We have now repeated the prognostic accuracy analyses in Figures 3B-D using only 8/13 ISGs, as suggested above. Although the *p*-values improved marginally, there was virtually no change in the ROC-AUC (Reviewer Figure Z3, see below). Further studies are likely needed in additional cohorts to identify the optimal subset of our 13-ISG panel for maximizing prognostic accuracy.

Reviewer Figure Z3. Metagene z-score of 8 type I IFN signaling genes in T, NK cells, B cells and myeloid cells. Boxplots of the pseudobulk (averaged across cells in one sample) metagene z-score in Progressors (n=11) vs Non-Progressors (n=52). *p*-value: Student’s *t*-test (greater). Receiver operating characteristic – area under curve (ROC-AUC) values show the accuracy of the 8-gene prognostic signature for predicting progression to more severe disease.

The paragraph starting line 288 does not make it clear that this analysis is using the same metagene score as earlier in the manuscript, but this time tracking expression by each individual over the disease course.

We have now updated the figure legends to indicate that the type I IFN metagene is normalized across baseline (Days 1-4) samples in **Figure 3B-D** (prognostic analysis) and across all samples in **Figure 4G-I** (longitudinal evolution).

While I understand that it can be useful to classify cells at an intermediate resolution for scRNA-seq, I believe that it is important to segregate out certain cell subsets that exhibit distinct dynamics to conventional T cells in viral infections, including COVID-19 e.g. MAIT cells.

We thank the reviewer for this suggestion. As described above, we have now repeated the T cell clonality statistics after excluding MAIT cells and confirmed that our conclusions remained the same (**Reviewer Figure Z1**, see above).

What is the prognostic score for IFN α protein relative to the 13 gene ISG signature (since you show a correlation in Fig. 3F)? IFN α would be easier to measure clinically than a gene signature.

We thank the reviewer for this suggestion. We have now updated **Figure 3G** with prognostic scores (ROC-AUC) for IFN α , MCP-1 and IP-10, as well as all three proteins combined. As indicated in the previous version of the manuscript, these three plasma proteins also have prognostic power for future worsening of COVID-19 severity.

Rather than testing the 33 Luminex proteins individually for differential expression at baseline between progressors and non-progressors, have the authors tested whether combinations of these proteins can predict disease progression, and how the accuracy/sensitivity of these protein signatures compares with the ISG metagene?

We agree with the reviewer that combinations of the 33 Luminex protein markers could potentially provide improved prognostic power relative to testing one marker at a time. However, since the number of such combinations is extremely large ($2^{33-34}=8,589,934,558$ non-trivial combinations in total), this approach would require a substantially larger cohort to achieve sufficient statistical power after correcting for multiple testing. As in the case of RNA markers, we therefore restricted our analysis to testing one marker at a time, as well as all significant markers together in a single panel.

Limitations regarding clinical use of the ISG signature as a biomarker of disease progression should be discussed.

We agree with the reviewer that clinical translation would be more straightforward with a smaller gene panel. We have now updated the Discussion section to indicate that:

“Analysis in larger cohorts would also provide the opportunity to identify the optimal subset of the 13 markers for clinical translation.”

Reviewer #2 (Remarks to the Author):

First of all, I appreciate the substantial additional work which the authors have made for the revision. Thanks to the extensive analyses and the related discussion, I consider the manuscript has

been very much improved. Especially, I appreciate the kind explanation on making use of the “pooled” samples. Thanks to their detailed explanation, I understand its reliable application. I also appreciate the extensive analyses on SOCS3. Even though some issues still remain to be further scrutinized, I understand that part should take time and would delay the publication of this manuscript in a timely manner, which I think is more important. Even now, there are a large number of patients who are newly infected or suffering from long-term symptoms. New viral strains are still emerging. I sincerely hope further in-depth analysis of the collected rich data should lead to better understandings of the mechanism of the viral infection

We are grateful to the reviewer for highlighting the value of this work, and of additional future analyses of this dataset once it is released.

Reviewer #4 (Remarks to the Author):

The revised version of the manuscript by Lin et al. is very interesting. The study is well-performed and the cohort of patients is quite wide collecting samples from early in disease. This work is supported by a large number of patients monitored throughout the disease course. The longitudinal study allows to determine possible biomarkers for prognostic purpose.

The responses provided to the Reviewer’s comments are satisfactory and have addressed the raised concerns; however, with regard to Response 2, I would suggest to add in the text that the cell state enrichment results for Mild 2 patients were not shown in Figure 2 because of the limited number of patients available to study.

We thank the reviewer for their positive comments regarding the study execution and the value of the early-stage cohort, as well as longitudinal profiling data.

We would like to clarify that cell state enrichment analysis of Mild2 samples was indeed performed, and the results consistently indicated upregulation of type I IFN genes in Progressor-enriched cell states. This analysis, originally shown in **Figure 2D-F**, was subsequently moved to **Supplementary Figure 6** in response to Reviewer 2’s comment that **Figure 2** was repetitive.

REVIEWERS' COMMENTS

Reviewer #1 (Remarks to the Author):

I have a few final comments/queries:

Was the batch variable (i.e. 20 batches) included in differential expression analyses between conditions? This would be important to include so that differences between conditions are not influenced by batch.

What ROC-AUC score do you get if you use the normalized expression of the 13 type I IFN genes rather than the z-score in Fig. 3B-D? This would be a more useful prognostic value since it is independent of the other samples present in the dataset.

“Z-scores can exaggerate differences between individuals. It would be helpful if the authors could include plots showing the absolute expression of each gene within the 13-gene signature in each individual, so that the reader can get a better idea of variability in expression of these genes across progressors and non-progressors.”

This previous comment was in relation to the patient cohort in this study rather than the German validation cohort.

Reviewer #1 (Remarks to the Author):

I have a few final comments/queries:

Was the batch variable (i.e. 20 batches) included in differential expression analyses between conditions? This would be important to include so that differences between conditions are not influenced by batch.

To reduce confounding by batch effects, we took care to ensure that all 20 batches had almost identical proportions of samples in each category of disease severity, gender, and ethnicity (see Line 152). Thus, we ensured that our results could not be confounded by cryptic correlations between disease severity and experimental batch in our dataset.

As mentioned in the Results text, we also reduced the effect of confounders in our study by balancing the age distributions of cases and controls and explicitly controlling for disease duration in all our analyses.

What ROC-AUC score do you get if you use the normalized expression of the 13 type I IFN genes rather than the z-score in Fig. 3B-D? This would be a more useful prognostic value since it is independent of the other samples present in the dataset.

As suggested, we have now recalculated the ROC-AUC scores in Figure 3B-D using normalized umi counts of the 13 prognostic genes, instead of z-scores. For all three cell lineages, in all three prognostic categories, the ROC-AUC scores remained in the same range as before (see below and compare to **Figure 3B-D**). We therefore retained the z-score-based AUC values in **Figure 3B-D**, for consistency with the rest of the manuscript.

Stage	NK, T cells	B cells	Myeloid
Mild1 Day 1-4	0.846	0.875	0.808
Mild1 Day 5-8	0.917	0.9	0.95
Mild2 Day 5-8	0.816	0.75	0.776

“Z-scores can exaggerate differences between individuals. It would be helpful if the authors could include plots showing the absolute expression of each gene within the 13-gene signature in each individual, so that the reader can get a better idea of variability in expression of these genes across progressors and non-progressors.”

This previous comment was in relation to the patient cohort in this study rather than the German validation cohort.

As suggested by the reviewer, we provide below a heatmap of the expression levels of the 13 genes without z-transformation. Since the 13 genes differ greatly in their average expression (~10-fold), this heatmap reveals gene-to-gene differences more clearly than it does donor-to-donor differences. In contrast, after z-transformation, expression variation across donors is more clearly visible in the heatmap, since all 13 genes span approximately the same color range (**Figure 2**).

NK, T cells

B cells

Myeloid